# Simple Minimax Optimal Byzantine Robust Algorithm for Nonconvex Objectives with Uniform Gradient Heterogeneity

**Tomoya Murata**[*]   **Kenta Niwa**[†]   **Takumi Fukami**[‡]   **Iifan Tyou**[§]

## Abstract

In this study, we consider nonconvex federated learning problems with the existence of Byzantine workers. We propose a new simple Byzantine robust algorithm called Momentum Screening. The algorithm is adaptive to the Byzantine fraction, i.e., all its hyperparameters do not depend on the number of Byzantine workers. We show that our method achieves the best optimization error of $O(\delta^2 \zeta_{\max}^2)$ for nonconvex smooth local objectives satisfying $\zeta_{\max}$-uniform gradient heterogeneity condition under $\delta$-Byzantine fraction, which can be better than the best known error rate of $O(\delta \zeta_{\mathrm{mean}}^2)$ for local objectives satisfying $\zeta_{\mathrm{mean}}$-mean heterogeneity condition when $\delta \leq (\zeta_{\mathrm{mean}}/\zeta_{\max})^2$. Furthermore, we derive an algorithm independent lower bound for local objectives satisfying $\zeta_{\max}$-uniform gradient heterogeneity condition and show the minimax optimality of our proposed method on this class. In numerical experiments, we validate the superiority of our method over the existing robust aggregation algorithms and verify our theoretical results.

## 1 Introduction

In distributed machine learning, some workers, called Byzantine workers (Lamport et al., 2019), may behave abnormally, so using the naive average aggregation of the workers' gradients to train a single global model can catastrophically degrade the model's accuracy. Examples of abnormal behavior include hardware crashes, message corruption, as well as the presence of poisoned data and the transmission of false information based on adversarial intent, and so on.

To tolerate the possibility of arbitrary malicious acts of Byzantine workers and to mitigate the degradation of the model accuracy, many robust aggregation rules have been proposed and analyzed in IID settings. Specifically each worker's dataset is randomly sampled from a common population (Blanchard et al., 2017; Yin et al., 2018; Alistarh et al., 2018; Guerraoui et al., 2018; Xie et al., 2019; Allen-Zhu et al., 2021; Karimireddy et al., 2021). Blanchard et al. (2017) theoretically studied Byzantine robustness for machine learning problems for the first time and analyzed KRUM, which is a distance-based robust aggregation. Yin et al. (2018) analyzed Coordinate Median (CM) and Trimmed Mean (TM) and derived statistically optimal rates with respect to the Byzantine fraction and the sample size, but they still explicitly depend on the problem dimension. Alistarh et al. (2018) developed a more sophisticated aggregation algorithm and derived an optimal error with a less dependence on the problem dimension than CM and TM, for convex cases. The algorithm has been further extended to nonconvex cases (Allen-Zhu et al., 2021). Karimireddy et al. (2021) theoretically showed that the existing historyless aggregations including KRUM, CM, and TM do not converge to the optimum for stochastic cases. In fact, they reported empirical failures of the historyless aggregations against time coupled attacks such as Inner Product Manipulation (IPM) (Xie et al., 2020) and A Little Is Enough (ALIE) (Baruch et al., 2019). To overcome this problem, they proposed a new aggregation called Centered Clipping (CClip) combined with momentum stochastic gradient descent (SGD) that exploits the history of the previous gradient information.

---

[*]`murata@msi.co.jp`, NTT DATA Mathematical Systems Inc.

[†]`kenta.niwa@ntt.com`, NTT Communication Science Laboratories, NTT Corporation

[‡]`takumi.fukami@ntt.com`, NTT Social Information Laboratories, NTT Corporation

[§]`iifan.tyou@ntt.com`, NTT Social Information Laboratories, NTT Corporation

In the progress of federated learning Konečnỳ et al. (2015); Shokri & Shmatikov (2015); McMahan et al. (2017), much attention has been given to the development of Byzantine robust algorithms, especially for non-IID cases, where the local datasets of the workers are heterogeneous. However, as shown in Li et al. (2023), the standard aggregations such as KRUM, CM, and TM empirically fail in non-IID cases due to the data heterogeneity. Many researchers have developed new robust aggregations for non-IID settings (Li et al., 2019; Data & Diggavi, 2021; Pillutla et al., 2022; Karimireddy et al., 2022; Allouah et al., 2023; Liu et al., 2023). Pillutla et al. (2022) proposed Robust Federated Averaging (RFA) based on Geometric Median (GM) aggregation for Federated Averaging (FedAvg), which is a standard optimizer in federated learning. In Data & Diggavi (2021), FedAvg is combined with a robust mean estimation algorithm. Karimireddy et al. (2022); Allouah et al. (2023); Liu et al. (2023) proposed a general robustness amplification technique for heterogeneous data.

Recent research on Byzantine robust algorithms can be categorized into three main directions: (i) asymptotic optimization accuracy improvement in non-IID cases, (ii) computational and communication efficiency improvement, and (iii) extensions of problem settings. In direction (i), Karimireddy et al. (2022) derived an algorithm independent optimization error lower bound and showed that no robust algorithms can reach the optimum in non-IID cases. Thus, it is crucial to investigate the best achievable optimization error of the robust algorithms in non-IID cases. Karimireddy et al. (2022) also gave a unified analysis of momentum SGD with a class of robust aggregators including KRUM, CM, and RFA wrapped by their proposed bucketing, and CClip and showed that these algorithms achieve a minimax optimal asymptotic optimization error for nonconvex local objectives satisfying the mean gradient heterogeneity condition introduced in Section 2 (Assumption 6). In direction (ii), many studies have proposed efficient robust algorithms by combining with variance reduction, multiple local updates, gradient compression and their combinations (Wu et al., 2020; Zhu & Ling, 2021; Data & Diggavi, 2021; Pillutla et al., 2022; Gorbunov et al.). In direction (iii), several extensions of the standard Byzantine robust optimization setting have been studied; learning on decentralized networks (Guo et al., 2020; El-Mhamdi et al., 2021; He et al., 2022), personalized federated learning (Li et al., 2021; Lin et al., 2022; Werner et al., 2023), combination with differential privacy (He et al., 2020; Guerraoui et al., 2021; Zhu & Ling, 2022), and so on.

This study particularly focuses on (i) optimization accuracy improvement for a specific class of nonconvex local objectives by developing a new simple robust algorithm in non-IID settings, because of its fundamental importance.

**Main Contributions.** We propose a new Byzantine robust algorithm called Momentum Screening (MS) for nonconvex federated learning. Its main features are as follows:

- (Algorithmic simplicity) MS uses a simple screening test to detect Byzantine workers and combines it with the standard momentum SGD.

- (Adaptivity to Byzantine fractions) Theoretically, MS does not need to know the number of Byzantine workers in advance, i.e., all hyperparameters of our algorithm are independent of Byzantine fraction $\delta \in [0, 0.5)$, which is highly desirable in practice.

- (Minimax optimality for $\mathcal{C}_{\mathrm{UH}}(\zeta_{\max})$) Thanks to the screening scheme, MS achieves the minimax optimal optimization error of $O(\delta^2 \zeta_{\max}^2)$ for local objectives class $\mathcal{C}_{\mathrm{UH}}(\zeta_{\max})$, that can be better than the previously known minimax optimal rate of $O(\delta \zeta_{\mathrm{mean}}^2)$ for class $\mathcal{C}_{\mathrm{MH}}(\zeta_{\mathrm{mean}})$[1].

To show the minimax optimality of MS, we derive not only an upper bound of the optimization error of MS, but also an algorithm independent lower bound for $\mathcal{C}_{\mathrm{UH}}(\zeta_{\max})$, and show that the obtained upper and lower bounds match in order sense. To the best of our knowledge, no lower bounds have been known for $\mathcal{C}_{\mathrm{UH}}(\zeta_{\max})$, and MS is the first minimax optimal algorithm for $\mathcal{C}_{\mathrm{UH}}(\zeta_{\max})$. The obtained rate has a better dependence on Byzantine fraction $\delta$ than previous ones, and thus MS can be more robust against Byzantine attacks for small $\delta$ on $\mathcal{C}_{\mathrm{UH}}(\zeta_{\max})$.

**Related Work.** Here, we discuss the connections between this study and the most relevant previous studies. Our screening idea is inspired by the pioneering work of Alistarh et al. (2018). However, the

---

[1]Roughly speaking, $\mathcal{C}_{\mathrm{UH}}(\zeta_{\max})$ denotes the class of nonconvex smooth local objectives satisfying that the *maximum* L2 distance of the local gradients from the global gradient is bounded by $\zeta_{\max}$. Similarly, $\mathcal{C}_{\mathrm{MH}}(\zeta_{\mathrm{mean}})$ denotes the class of nonconvex smooth local objectives satisfying that the *mean* L2 distance of the local gradients from the global gradient is bounded by $\zeta_{\mathrm{mean}}$. For their formal definitions, see Definition 1 in Section 2.

algorithm presented in Alistarh et al. (2018) is more complex than ours because it does not rely on the momentum technique and needs to aggregate three variables, and their problem setting is limited to IID cases, which are not standard in federated learning. Although the objective functions considered in Alistarh et al. (2018) are convex, the algorithm and the analysis have been extended to nonconvex cases (Allen-Zhu et al., 2021). CClip was first proposed in the context of IID settings (Karimireddy et al., 2021). After that, Karimireddy et al. (2022) gave a unified analysis of momentum SGD with their so-called robust aggregators, which includes KRUM, CM, and RFA wrapped by bucketing and CClip on non-IID settings. CClip clips the difference in each input from some guess of the true mean and efficiently mitigates the Byzantine behavior even for heterogeneous local objectives. Bucketing first makes a random partition of input variables, takes the average on each bucket, and then applies traditional robust aggregations to the averaged results. Karimireddy et al. (2021) showed that these algorithms achieve the minimax optimal rate $O(\delta \zeta_{\mathrm{mean}}^2)$ for local objectives satisfying $\zeta_{\mathrm{mean}}$-mean gradient heterogeneity condition (Assumption 6). In contrast, this study focuses on another local objectives class satisfying $\zeta_{\mathrm{max}}$-*uniform gradient heterogeneity condition (Assumption 5)* and derives the minimax optimal rate $O(\delta^2 \zeta_{\mathrm{max}}^2)$ of our method, which implies that our method achieves better optimization accuracy than CClip and bucketing when $\delta \leq (\zeta_{\mathrm{mean}}/\zeta_{\mathrm{max}})^2$ holds[2]. Also, CClip and bucketing are not adaptive to Byzantine fraction $\delta$, which means that the theoretically justified hyperparameters of the algorithms depend on the true $\delta$. In contrast, our method is adaptive to $\delta$. Very recently, Allouah et al. (2023) proposed a general robustness amplification technique called Nearest Neighbor Mixing (NNM), which first creates new input variables by mixing each input point with its nearest neighbor and then feeds the results into standard robust aggregation algorithms. This increases the aggregation accuracy, and NNM has been shown to achieve optimization error of $O(\delta \zeta_{\mathrm{mean}}^2)$, which is the same as that of CClip and bucketing, for $\zeta_{\mathrm{mean}}$-mean gradient heterogeneous local objectives. Also, very recently Liu et al. (2023) has proposed another general robustness amplification called GrAdient Splitting (GAS), which first splits the coordinates into $p$ subsets and then applies standard robust aggregations to the input variables on each split coordinate. GAS with splitting size $p$ has been shown to achieve optimization error of $O((\delta^2(1 + |\mathcal{G}|/p)(\zeta_{\mathrm{max}}^2 + \sigma^2))$ for $\zeta_{\mathrm{max}}$-uniform gradient heterogeneous local objectives, which is worse than our obtained rate due to the additional dependence on the number of honest workers $|\mathcal{G}|$ and stochastic gradient variance $\sigma^2$. Also, neither NNM nor GAS is adaptive to $\delta$.

## 2 NOTATION AND PROBLEM SETTINGS

In this section, some notations used in this paper are introduced. Then, our problem settings and theoretical assumptions are given.

**Notation.** $\| \cdot \|$ denotes the Euclidean $L_2$ norm $\| \cdot \|_2$: $\|x\| = \sqrt{\sum_i x_i^2}$ for vector $x$. For a matrix $X$, $\|X\|$ denotes the induced norm by the Euclidean $L_2$ norm. For a natural number $m$, $[m]$ denotes the set $\{1, 2, \ldots, m\}$. For a finite set $A$, $|A|$ denotes the number of elements. For any number $a, b$, $a \vee b$ denotes $\max\{a, b\}$ and $a \wedge b$ denotes $\min\{a, b\}$.

**Problem Settings.** Let $n$ be the number of workers. We assume that $\delta n$ workers can be Byzantine ($\delta \in [0, 0.5)$) and can send arbitrary vectors to the central server in the training. Let $\mathcal{G}$ be the set of non-Byzantine workers ($|\mathcal{G}| = (1 - \delta)n$). We want to minimize nonconvex objective $f(x) := \frac{1}{|\mathcal{G}|} \sum_{i \in \mathcal{G}} f_i(x)$, where $f_i(x) := \mathbb{E}_{z \sim \mathcal{D}_i}[\ell(x, z)]$. Here, $\mathcal{D}_i$ denotes the local data distribution of worker $i$. Since the objective function is nonconvex, we set our goal to find approximate first-order optimal points $x$, i.e., $\|\nabla f(x)\|^2 \leq \varepsilon$ given the desired optimization error $\varepsilon > 0$. It is not assumed that $\mathcal{D}_i = \mathcal{D}_j$ for $i \neq j$ and thus the local data distributions can be *heterogeneous*. In this case, we can no longer generally expect the asymptotic optimization error $\lim_{t \to \infty} \|\nabla f(x_t)\|^2$ to be zero in the presence of Byzantine workers, due to the bias from the data heterogeneity (Karimireddy et al., 2022). Therefore, we focus on finding $x$ that satisfies $\|\nabla f(x)\|^2 \leq \varepsilon$ for as small $\varepsilon > 0$ as possible.

### 2.1 ASSUMPTIONS

In this subsection, several assumptions used in Section 4 are introduced.

**Assumption 1.** $f_i$ *is L-smooth for each* $i \in \mathcal{G}$.

---

[2]For the discussions of this condition, see Sections 2 and 7.

**Assumption 2.** *There exists a global minima $x_*$ of $f$.*

**Assumption 3.** *Minibatch stochastic gradient $g_i$ at $x$ of $f_i$, with $\mathbb{E}[g_i] = \nabla f_i(x)$, satisfies the following norm sub-Gaussian property with parameter $\sigma^2$ for every $x \in \mathbb{R}^d$ and $i \in \mathcal{G}$: $\mathbb{P}_{z \sim \mathcal{D}_i}(\|g_i - \nabla f_i(x)\| \geq s) \leq 2 \exp(-s^2/(2\sigma^2))$ for any $s \geq 0$.*

Assumption 3 is necessary to obtain high probability bounds and is often assumed in the stochastic optimization literature (Jin et al., 2021).

**Assumption 4.** *$\|\nabla \ell(x, z)\| \leq G$ holds for every $x \in \mathbb{R}^d$ a.s. with respect to $z \sim \mathcal{D}_i$ for each $i \in \mathcal{G}$.*

Assumption 4 is only used in the application of Azuma-Hoeffding's inequality for norm-subGaussian martingales (Jin et al., 2019). Very importantly, *G depends only log log order on the iteration complexity of our algorithm to achieve the best optimization error* and *G never depends on the final optimization error* in our theory (see Theorem 1 in Section 4). Hence, Assumption 4 is not so limited.

**Assumption 5.** *$\{f_i\}_{i \in \mathcal{G}}$ satisfies the following $\zeta_{\max}$-uniform gradient heterogeneity condition:*

$$\max_{i \in \mathcal{G}} \|\nabla f_i(x) - \nabla f(x)\|^2 \leq \zeta_{\max}^2, \forall x \in \mathbb{R}^d.$$

**Assumption 6.** *$\{f_i\}_{i \in \mathcal{G}}$ satisfies the following $\zeta_{\text{mean}}$-mean gradient heterogeneity condition:*

$$\frac{1}{|\mathcal{G}|} \sum_{i \in \mathcal{G}} \|\nabla f_i(x) - \nabla f(x)\|^2 \leq \zeta_{\text{mean}}^2, \forall x \in \mathbb{R}^d.$$

Note that both Assumptions 5 and 6 measure some heterogeneity of the non-Byzantine local objectives $\{f_i\}_{i \in \mathcal{G}}$. Assumption 5 is stronger than Assumption 6, because $\zeta_{\max} \geq \zeta_{\text{mean}}$ always holds. In general, the relationship between $\zeta_{\text{mean}}$ and $\zeta_{\max}$ depends on the structure of the local datasets. For example, consider learning a model on CIFAR10 dataset with 20 workers in two scenarios. In the first scenario, workers 1 through 19 have a local dataset randomly sampled from CIFAR10 in an IID manner, and worker 20 has only a local dataset consisting of random samples with class label 0. Then we expect that $\zeta_{\max}/\zeta_{\text{mean}} \gg 1$ due to the presence of outlier worker 20. In the second scenario, each worker has a local dataset consisting of random samples from two random classes in $\{0, \ldots, 9\}$. In this case, we expect $\zeta_{\max}/\zeta_{\text{mean}}$ to be much smaller than in the first scenario, even though the local datasets are still highly heterogeneous. Thus, *the quantity $\zeta_{\max}/\zeta_{\text{mean}}$ describes a property of the heterogeneity rather than its strength*. We empirically observed that for several neural networks, including VGG11, $\zeta_{\max}/\zeta_{\text{mean}}$ was not so large (roughly $1.0 \sim 3.0$) on MNIST and CIFAR10 with heterogeneous allocation and even on a realistic federated learning dataset (see Section E).

**Definition 1.** *$\mathcal{C}_{\text{UH}}(\zeta_{\max})$ is defined as the set of local objectives $\{f_i\}_{i \in \mathcal{G}}$ that satisfies Assumptions 1, 2, 3, 4, and 5. Similarly, $\mathcal{C}_{\text{MH}}(\zeta_{\text{mean}})$ is defined as the set of local objectives $\{f_i\}_{i \in \mathcal{G}}$ that satisfies Assumptions 1, 2, 3, 4 and 6.*

Observe that $\mathcal{C}_{\text{UH}}(\zeta) \subset \mathcal{C}_{\text{MH}}(\zeta)$. Most previous work has essentially focused on $\mathcal{C}_{\text{MH}}(\zeta_{\text{mean}})$. In our theory, *we focus on $\mathcal{C}_{\text{UH}}(\zeta_{\max})$ to improve the asymptotic optimization error obtained so far*.

## 3 APPROACH AND PROPOSED ALGORITHM

In this section, we illustrate our ideas and the proposed algorithm.

**Review of Centered Clipping.** The algorithm most closely related to this work is Centered Clipping (CClip) Karimireddy et al. (2021; 2022), which is a state-of-the-art robust aggregation algorithm that empirically works much better than the traditional ones like KRUM, CM, and RFA for Byzantine robust federated learning. The simplest CClip takes independent random vectors $\{m_i^t\}_{i=1}^n$ as input and returns $v + (1/n) \sum_{i=1}^n \min\{1, \tau/\|m_i^t - v\|\}(m_i^t - v)$, where $v$ is an initial guess of the ideal aggregation $(1/|\mathcal{G}|) \sum_{i \in \mathcal{G}} m_i^t$, and is often set to the previous aggregation result $m^{t-1}$, but we repeatedly apply CClip by replacing the initial guess with the current aggregation $m^t$. The clipping radius $\tau$ is derived theoretically as $\tau = \Theta(\rho/\sqrt{\delta})$, where $\rho := \max_{i,j \in \mathcal{G}} \sqrt{\mathbb{E}\|m_i^t - m_j^t\|^2}$ and $\delta \leq 0.1$ is the Byzantine fraction. A key observation of CClip's update rule is that even for the theoretically determined $\tau$, the individual component $m_i^t$ may be clipped and biased toward zero for non-Byzantine worker $i$. This means that CClip is conservative in a sense: one can mitigate the effect

---

**Algorithm 1:** Momentum Screening($x^0, \eta, \alpha, \{\tau_t\}_{t=1}^T$)

---

1: **for** $t = 1$ to $T$ **do**
2:     **for** $i \in [n]$ in parallel **do**
3:         **if** $i \in \mathcal{G}$ **then**
4:            Compute minibatch stochastic gradients $g_i^t$ at $x^{t-1}$.
5:            Send $m_i^t = (1 - \alpha)m_i^{t-1} + \alpha g_i^t$ ($m_i^0 = g_i^1$) to the server.
6:         **else**
7:            Send arbitrary vector $\in \mathbb{R}^d$ to the server. # Worker $i$ is Byzantine.
8:         **end if**
9:     **end for**
10:    $m^t = \text{Screen}(\{m_i^t\}_{i=1}^n, \tau_t)$.
11:    $x^t = x^{t-1} - \eta m^t$.
12: **end for**
13: **Return:** $x^{\hat{t}}$ ($\hat{t} \sim \text{Unif}[T]$).

---

**Algorithm 2:** Screen($\{m_i\}_{i=1}^n, \tau$)

---

1: $\hat{\mathcal{G}} = \{i \in [n] : |\{j \in [n] : \|m_i - m_j\| \leq \tau\}| \geq 0.5n\}$.
2: $m := \frac{1}{|\hat{\mathcal{G}}|} \sum_{i \in \hat{\mathcal{G}}} m_i$
3: **Return:** $m$.

---

of adversarial variables at the expense of the clipping bias for non-Byzantine workers. For the more details of CClip, see Karimireddy et al. (2021; 2022).

**Our Approach: Momentum Screening.** One feature of CClip is to allow clipping bias for non-Byzantine workers. Another possible approach is to use a safe detection test to remove adversarial variables $\{m_i^t\}_{i \in [n] \setminus \mathcal{G}}$, where all the non-Byzantine workers must pass the detection test, and hopefully many more Byzantine workers will be detected by the test and their variables screened out (i.e., removed). If this type of detection test is available, the non-Byzantine variables will not be affected by the aggregation. In this case, the aggregation error only comes from the bias of the adversarial vectors that could not be detected by the test. To realize such a detection test, we consider a simple *screening* approach: an input vector $m_i^t$ is screened out (i.e., removed) if the $d$-dimensional sphere with center $m_i^t$ and radius $\tau_t$ does not contain at least half of the $\{m_i^t\}_{i=1}^n$. A similar idea to screening already appeared in Alistarh et al. (2018), but the paper only gave a theoretical analysis in the context of homogeneous settings that are not standard in federated learning, and importantly the whole algorithm is very complex because their algorithm aggregates three quantities (i) the current gradients; (ii) the accumulated gradients; and (iii) the accumulated inner products between the previous gradients and the updated differences, and screening is applied to both (i) and (ii). In contrast, we simply apply screening to the momentum $\{m_i\}_{i=1}^n$ only once per round.

**Concrete Procedures.** The concrete procedures of our proposed algorithm Momentum Screening (MS) are described in Algorithm 1. For each iteration $t$, each worker computes a minibatch stochastic gradient $g_i^t$ and updates the momentum $m_i^t$ in parallel. Then, the central server receives the updated momentums $\{m_i^t\}_{i=1}^n$ and aggregates them using Screen (Algorithm 2). The aggregated momentum is then used to update the global model parameter. In Algorithm 2, we screen "malicious" momentum $m_i$ by measuring the distance to the other momentums $m_j$; if the sphere with center $m_i$ and radius $\tau_t$ contains only less than half of the other momentums, $m_i$ is judged to be anomalous and decided to be rejected. Then, the output is the average of the momentums that pass this screen test. As shown in Section 4, it is theoretically nice to set $\tau_t$ to $\Theta(\zeta_{\max}) + \widetilde{\Theta}((1 - \Omega(\alpha))^{t-1} + \alpha)^{0.5}\sigma$ to avoid removing the output of non-Byzantine workers, where $\alpha$ is the momentum parameter. In practice, we use some decay scheduling for $\{\tau_t\}_{t=1}^T$. For our experimental settings of $\{\tau_t\}_{t=1}^T$, see Section 6 and D.2.

**Remark 1** (Efficient implementation of Algorithm 2)**.** *A naive implementation of Algorithm 2 requires $O(n^2 d)$ computational cost[3], which is worse than $\widetilde{O}(nd)$ of CClip and sometimes forbidden. However, we can efficiently implement Algorithm 2 with only $\widetilde{O}(nd)$ expected computational cost. The*

---

[3]This is because we need to check $\|m_i - m_j\| \leq \tau$ for every $i, j \in [n]$.

*concrete procedure of the efficient implementation of Algorithm 2 is given in Algorithm 3 in Section C due to space limitations. We can show that the aggregation error $\|m^t - \bar{m}^t\|$ with Algorithm 3 is only nearly $1.5$ times worse than that with Algorithm 2, where $\bar{m}^t := (1/|\mathcal{G}|) \sum_{i \in \mathcal{G}} m_i^t$ means the ideal aggregation result. For more details, see Proposition 2 in Section 4. Also, we empirically observed that the performance of Algorithm 1 with Algorithm 3 was similar to that with 2 (the empirical comparison can be found in Section D.5).*

## 4 CONVERGENCE ANALYSIS

In this section, we provide a theoretical analysis of Algorithm 1. $\widetilde{O}$ and $\widetilde{\Theta}$ abbreviate extra poly-logarithmic factors depending on $d$, $1/q$, $T$, $\log |\mathcal{G}|$, $\log(1/\sigma^2)$, and $\log G$ for simple presentation, where $q \in (0, 1)$ is the confidence parameter of a high probability bound. All the proofs can be found in Section A.

**Overview of Analysis.** Lemma 1 and Proposition 1 give the standard bounds for stochastic gradient methods with momentum. These are very similar to Lemmas 9 and 10 in Karimireddy et al. (2022), but our results give *high probability bounds* instead of the expectation bounds proved in Karimireddy et al. (2022). This is important for the analysis of our screening algorithm because evaluating the aggregation error $\|m_i^t - m_j^t\|^2$ in expectation is difficult since Algorithm 2 relies on the observed value of diameter $\|m_i - m_j\|$ without expectation. To derive high probability bounds, we use Azuma-Hoeffding's inequality for norm-subGaussian martingales (Jin et al., 2019). Proposition 2 gives the aggregation error bound $O(\delta^2 \rho_{\max}^2)$ for Algorithm 2 with appropriate $\tau = \Theta(\rho_{\max})$ under the uniform diameter bound $\max_{i,j \in \mathcal{G}} \|m_i^t - m_j^t\| \leq \rho_{\max}$, which is better than that of Karimireddy et al. (2022) thanks to the screening scheme, and this part is the most important in our analysis. Proposition 3 gives a high probability bound for the diameter $\|m_i^t - m_j^t\|$, which is required in Proposition 2 and roughly we show that $\max_{i,j \in \mathcal{G}} \|m_i^t - m_j^t\| \leq O(\zeta_{\max}) + \widetilde{O}(((1-\alpha)^{t-1} + \alpha)\sigma^2)$ by essentially using Assumption 5. This means that the theoretically justified $\tau_t$ is $\Theta(\zeta_{\max}) + \widetilde{\Theta}(((1-\alpha)^{t-1} + \alpha)\sigma^2)$ and thus *the setting of $\tau_t$ is adaptive to the Byzantine fraction $\delta$*. In Theorem 1, we combine all the results and derive a convergence rate of Algorithm 1 and show an asymptotic optimization error of $(1/T) \sum_{t=1}^{T} \|\nabla f(x^{t-1})\|^2 \leq O(\delta^2 \zeta_{\max}^2)$, which can be better than that of $O(\delta \zeta_{\mean}^2)$ in the previous studies.

**Lemma 1** (Descent Lemma). *Suppose that Assumption 1 holds. Let $\alpha \in (0, 1]$, $\eta \leq 1/L$. Then, for every $t \geq 1$ it holds that $f(x^t) \leq f(x^{t-1}) - (\eta/2)\|\nabla f(x^{t-1})\|^2 + \eta \|\bar{e}^t\|^2 + \eta \|m^t - \bar{m}^t\|^2$. Here, $\bar{e}^t := \bar{m}^t - \nabla f(x^{t-1})$ and $\bar{m}^t := (1/|\mathcal{G}|) \sum_{i \in \mathcal{G}} m_i^t$.*

**Proposition 1.** *Suppose that Assumptions 1, 3 and 4 hold. Let $q \in (0, 1/2)$, $\alpha \in (0, 1)$ and $\eta \leq \sqrt{\alpha/(48(1 + 1/\alpha))}(1/L) = \Theta(\alpha/L)$. Then, it holds that for fixed $t \geq 1$*

$$\|\bar{e}^t\|^2 \leq A \sum_{\tau=2}^{t} \left(1 - \frac{\alpha}{4}\right)^{t-\tau} \|m^{\tau-1} - \bar{m}^{\tau-1}\|^2 + B \sum_{\tau=2}^{t} \left(1 - \frac{\alpha}{4}\right)^{t-\tau} \|\nabla f(x^{\tau-2})\|^2 + C_t$$

*with probability at least $1 - 2q$. Here, $A = O(\eta^2 L^2/\alpha)$, $B := \alpha(1/16 + 4c\gamma(1 - \alpha))$ and $C_t = \widetilde{O}(((1 - \alpha/4)^{t-1} + \alpha/\gamma)\sigma^2/|\mathcal{G}|))$, where $\gamma := 1/(128c(1 - \alpha))$ and $c$ is some universal constant.*

**Proposition 2** (Aggregation Error Bound). *Suppose that Assumption 5 holds. Let $t \in [T]$. If $\|m_i^t - m_j^t\|^2 \leq \rho_{\max}^2$ for every $i, j \in \mathcal{G}$, under $\tau_t \geq \rho_{\max}$, Algorithm 2 satisfies $\|m^t - \bar{m}^t\|^2 \leq 4\delta^2 \tau_t^2$, and for $q \in (0, 1)$, with probability at least $1 - q$, Algorithm 3 satisfies $\|m^t - \bar{m}^t\|^2 \leq 4(1 + s)^2 \delta^2 \tau_t^2$, where $s \geq \delta + 3\sqrt{(\log(2/q))/(2K)}$ is a parameter of Algorithm 3.*

**Proposition 3** (Diameter Bound). *Let $q \in (0, 1/2)$ and $\alpha \in (0, 1/2)$. Suppose that Assumptions 3, 4 and 5 hold. Then, it holds that for fixed $t \geq 1$, with probability at least $1 - 2q$, for any $i, j \in \mathcal{G}$,*

$$\|m_i^t - m_j^t\|^2 \leq O(\zeta_{\max}^2) + \widetilde{O}\left(\left(\left(1 - \frac{\alpha}{4}\right)^{t-1} + \alpha\right)\sigma^2\right).$$

**Theorem 1.** *Suppose that Assumptions 1, 2, 3, 4 and 5 hold. Let $\eta \leq 1/(8\sqrt{6}L)$, $\alpha := 4\sqrt{6}\eta L (\leq 1/2)$ and $\tau_t = \Theta(\zeta_{\max}^2) + \widetilde{\Theta}(((1-\alpha)^{t-1} + \alpha)\sigma^2)$ be appropriately chosen. Algorithm 1 satisfies*

$$\frac{1}{T} \sum_{t=1}^{T} \|\nabla f(x^{t-1})\|^2 \leq O\left(\frac{f(x^0) - f(x_*)}{\eta T}\right) + O\left(\delta^2 \zeta_{\max}^2\right) + \widetilde{O}\left(\left(\frac{1}{\eta L T} + \eta L\right)\left(\delta^2 + \frac{1}{|\mathcal{G}|}\right)\sigma^2\right)$$

*with probability at least $1 - 5q$ for any $q \in (0, 1/5)$. In particular, if we set $\eta := (1/(8\sqrt{6}L)) \wedge (1/(\sqrt{T}L))$, for sufficiently large $T$ we obtain $(1/T)\sum_{t=1}^{T} \|\nabla f(x^{t-1})\|^2 \leq O(\delta^2 \zeta_{\max}^2)$.*

**Remark 2.** *The best achievable optimization error $O(\delta^2 \zeta_{\max}^2)$ can be better than the state-of-the-art one $O(\delta \zeta_{\mean}^2)$ of CClip and bucketing (Karimireddy et al., 2022) when $\delta \leq (\zeta_{\mean}/\zeta_{\max})^2$.*

**Remark 3.** *From Theorem 1, we can confirm that the hyperparameters $\eta$, $\alpha$ and $\tau$ are independent of $\delta$. In this sense, our algorithm is adaptive to Byzantine fraction $\delta$.*

## 5 LOWER BOUND FOR $\mathcal{C}_{\mathrm{UH}}(\zeta_{\max})$

In this section, we discuss a lower bound for $\mathcal{C}_{\mathrm{UH}}(\zeta_{\max})$ and the optimality of MS (Algorithm 1).

The basic approach is based on Karimireddy et al. (2022). However, for the construction of algorithm-independent worst-case local objectives contained in $\mathcal{C}_{\mathrm{UH}}(\zeta_{\max})$, we need to carefully construct quadratic functions on $\mathbb{R}$ and rely on their smoothed versions, which is not necessary for the analysis of Karimireddy et al. (2022). The following theorem gives a lower bound $\Omega(\delta^2 \zeta_{\max}^2)$ for $\mathcal{C}_{\mathrm{UH}}(\zeta_{\max})$ and combining the lower bound with Theorem 1 implies the minimax optimality of Algorithm 1 in terms of asymptotic optimization error.

**Theorem 2.** *Let $\delta \in [0, 0.5)$. For any optimization algorithm $\mathcal{A}$, there exists a sequence of local objectives $\{f_i\}_{i \in [(1-\delta)n]} \in \mathcal{C}_{\mathrm{UH}}(\zeta_{\max})$ such that for objective function $f(x) := (1/|\mathcal{G}|)\sum_{i \in \mathcal{G}} f_i(x)$ with $\mathcal{G} := [(1 - \delta)n]$ it holds that $\mathbb{E}_\pi \|\nabla f(\mathcal{A}(\{f_{\pi(i)}\}_{i=1}^n))\|^2 \geq \Omega(\delta^2 \zeta_{\max}^2)$. Here, $\pi$ is a random permutation over $[n]$ and the expectation is taken w.r.t. the randomness of $\pi$.*

**Remark 4.** *In Karimireddy et al. (2022), a lower bound $O(\delta \zeta_{\mean}^2)$ is essentially derived for function class $\mathcal{C}_{\mathrm{MH}}(\zeta_{\mean})$. Theorem 2 does not contradict this lower bound because Theorem 2 gives the lower bound for $\mathcal{C}_{\mathrm{UH}}(\zeta_{\max})$ rather than $\mathcal{C}_{\mathrm{MH}}(\zeta_{\mean})$.*

## 6 NUMERICAL EXPERIMENTS

In this section, we provide the results of our numerical experiments to verify that our algorithm is superior to the previous algorithms. [4]

**Data preparation.** We used two standard 10-class classification datasets MNIST[5] and CIFAR10[6]. In our experiments, we set the total number of workers to $n = 20$ and the number of non-Byzantine workers to $20(1 - \delta)$ for Byzantine fraction $\delta$. To construct $20(1 - \delta)$ heterogeneous local datasets, we adopted the procedures used in Karimireddy et al. (2022). First, the training dataset was randomly divided into 50%. Next, one 50% dataset was equally divided and distributed to each non-Byzantine worker. Finally, the other 50% dataset was sorted by labels and divided sequentially into 20 equal parts[7] and distributed to each non-Byzantine worker. This means that each non-Byzantine worker had 50% IID samples and 50% highly non-IID samples. These procedures were done independently for MNIST and CIFAR10.

**Models.** Our experiments were conducted using three neural newtorks; (i) a one-hidden layer fully connected neural network (FC), (ii) a convolutional neural network with two convolutional layers and two fully connected layers (CNN), and (iii) VGG11 (Simonyan & Zisserman, 2014). The details of the network architectures are found in Section D.1. Due to space limitations, we only report the results on (i) FC and (iii) VGG11, and the results on (ii) CNN can be found in Section D.4.

**Implemented aggregation algorithms.** We implemented six aggregation algorithms: simple averaging (Avg), Coordinate Median (CM), KRUM (Blanchard et al., 2017), RFA (Pillutla et al., 2022), Centered Clipping (CClip) (Karimireddy et al., 2022) and our proposed Momentum Screening (MS, Algorithm 1 with Algorithm 2). The details of them are found in Section D.2. We fixed learning rate $\eta$ to 0.01, minibatch size to 32. We set the number of epochs to 20 for FC and CNN, and 100 for VGG11. Also, we decided to apply a momentum technique with $\alpha = 0.1$ for every algorithm because

---

[4]The source code of our experiment will be publicly available after the paper is published.

[5]http://yann.lecun.com/exdb/mnist/.

[6]https://www.cs.toronto.edu/~kriz/cifar.html.

[7]From the construction procedures, we can expect that $\zeta_{\max}/\zeta_{\mean}$ will not be so large, because each local dataset was treated approximately equally in a sense, and there were no outlier local datasets.

| Model/Data | AGG | BF | LF | Mimic | IPM | ALIE | Worst |
|---|---|---|---|---|---|---|---|
| FC/ MNIST | Avg | $95.1 \pm 0.2$ | $\mathbf{95.5 \pm 0.3}$ | $\mathbf{95.5 \pm 0.3}$ | $\mathbf{94.8 \pm 0.1}$ | $89.3 \pm 0.7$ | $89.3 \pm 0.7$ |
|  | CM | $93.1 \pm 0.6$ | $93.3 \pm 0.2$ | $94.1 \pm 0.6$ | $91.4 \pm 0.6$ | $88.2 \pm 3.2$ | $88.2 \pm 3.2$ |
|  | KRUM | $93.0 \pm 0.3$ | $94.0 \pm 0.4$ | $94.5 \pm 1.0$ | $92.8 \pm 0.4$ | $\mathbf{95.1 \pm 0.1}$ | $92.8 \pm 0.3$ |
|  | RFA | $94.7 \pm 0.2$ | $95.3 \pm 0.3$ | $95.3 \pm 0.4$ | $93.7 \pm 0.2$ | $90.2 \pm 0.5$ | $90.2 \pm 0.5$ |
|  | CClip | $94.8 \pm 0.2$ | $95.2 \pm 0.3$ | $95.4 \pm 0.3$ | $93.7 \pm 0.2$ | $93.2 \pm 0.4$ | $93.2 \pm 0.4$ |
|  | MS (ours) | $\mathbf{95.2 \pm 0.2}$ | $95.4 \pm 0.3$ | $\mathbf{95.5 \pm 0.3}$ | $94.5 \pm 0.1$ | $94.9 \pm 0.2$ | $\mathbf{94.5 \pm 0.1}$ |
| VGG11/ MNIST | Avg | $\mathbf{99.3 \pm 0.1}$ | $\mathbf{99.3 \pm 0.1}$ | $\mathbf{99.4 \pm 0.1}$ | $\mathbf{99.3 \pm 0.1}$ | $30.8 \pm 15.1$ | $30.8 \pm 15.1$ |
|  | CM | $99.2 \pm 0.1$ | $99.1 \pm 0.1$ | $99.3 \pm 0.1$ | $99.1 \pm 0.0$ | $67.0 \pm 10.5$ | $67.0 \pm 10.5$ |
|  | KRUM | $98.9 \pm 0.1$ | $99.2 \pm 0.1$ | $99.0 \pm 0.1$ | $98.7 \pm 0.1$ | $99.2 \pm 0.1$ | $98.7 \pm 0.1$ |
|  | RFA | $\mathbf{99.3 \pm 0.1}$ | $\mathbf{99.3 \pm 0.1}$ | $99.3 \pm 0.1$ | $\mathbf{99.3 \pm 0.1}$ | $72.8 \pm 34.7$ | $72.8 \pm 34.7$ |
|  | CClip | $\mathbf{99.3 \pm 0.1}$ | $\mathbf{99.3 \pm 0.1}$ | $99.3 \pm 0.1$ | $\mathbf{99.3 \pm 0.1}$ | $95.3 \pm 2.8$ | $95.3 \pm 2.8$ |
|  | MS (ours) | $\mathbf{99.3 \pm 0.1}$ | $\mathbf{99.3 \pm 0.0}$ | $99.3 \pm 0.1$ | $99.0 \pm 0.3$ | $\mathbf{99.3 \pm 0.0}$ | $\mathbf{99.0 \pm 0.3}$ |
| FC/ CIFAR10 | Avg | $\mathbf{46.7 \pm 1.3}$ | $\mathbf{46.9 \pm 1.4}$ | $\mathbf{46.1 \pm 1.2}$ | $\mathbf{46.7 \pm 1.3}$ | $25.2 \pm 3.3$ | $25.2 \pm 3.3$ |
|  | CM | $39.6 \pm 2.2$ | $39.6 \pm 0.9$ | $40.2 \pm 1.6$ | $37.6 \pm 1.3$ | $27.4 \pm 1.7$ | $27.4 \pm 1.7$ |
|  | KRUM | $35.6 \pm 1.9$ | $38.6 \pm 1.2$ | $38.2 \pm 3.4$ | $33.3 \pm 1.4$ | $37.7 \pm 2.5$ | $33.7 \pm 2.1$ |
|  | RFA | $46.2 \pm 0.7$ | $46.7 \pm 0.8$ | $45.9 \pm 2.0$ | $45.8 \pm 1.0$ | $29.0 \pm 3.7$ | $29.0 \pm 3.7$ |
|  | CClip | $44.5 \pm 1.2$ | $45.7 \pm 0.6$ | $44.0 \pm 3.5$ | $40.9 \pm 1.0$ | $35.4 \pm 0.8$ | $35.4 \pm 0.8$ |
|  | MS (ours) | $46.3 \pm 1.1$ | $46.2 \pm 1.3$ | $45.2 \pm 1.6$ | $45.8 \pm 1.9$ | $\mathbf{45.0 \pm 2.5}$ | $\mathbf{44.6 \pm 2.0}$ |
| VGG11/ CIFAR10 | Avg | $\mathbf{84.3 \pm 0.9}$ | $\mathbf{85.0 \pm 0.4}$ | $\mathbf{85.1 \pm 0.8}$ | $\mathbf{84.5 \pm 0.3}$ | $19.2 \pm 1.3$ | $19.2 \pm 1.3$ |
|  | CM | $45.6 \pm 2.5$ | $43.7 \pm 4.3$ | $57.2 \pm 9.2$ | $34.9 \pm 3.7$ | $19.1 \pm 1.9$ | $19.1 \pm 1.9$ |
|  | KRUM | $55.8 \pm 2.5$ | $64.2 \pm 1.8$ | $70.3 \pm 2.2$ | $40.6 \pm 4.8$ | $71.9 \pm 8.3$ | $40.6 \pm 4.8$ |
|  | RFA | $82.7 \pm 0.3$ | $83.9 \pm 0.2$ | $84.2 \pm 0.4$ | $81.5 \pm 0.6$ | $20.3 \pm 1.3$ | $20.3 \pm 1.3$ |
|  | CClip | $77.9 \pm 0.7$ | $81.3 \pm 0.6$ | $81.3 \pm 0.6$ | $64.2 \pm 18.3$ | $22.7 \pm 2.3$ | $22.7 \pm 2.3$ |
|  | MS (ours) | $84.2 \pm 0.4$ | $84.6 \pm 0.6$ | $84.8 \pm 0.9$ | $83.5 \pm 0.8$ | $\mathbf{83.3 \pm 3.4}$ | $\mathbf{82.8 \pm 2.5}$ |

Table 1: Comparison of 95% confidence intervals of the best test accuracy (%) against five attacks ($\delta = 3/20$) for each model and dataset ("Worst" shows the confidence interval of the *worst* best test accuracy among five attacks). Bucketing technique was combined with CM, KRUM, and RFA.

it consistently improved the optimization accuracy in our experiments[8]. For CM, KRUM, and RFA, we tested the aggregation with the bucketing technique. In contrast, the bucketing technique was not applied to CClip and MS[9]. The bucketing parameter $s$ was set to 2 in accordance with Karimireddy et al. (2022). For the clipping parameter of CClip, we followed the suggested parameter described in Karimireddy et al. (2022). For MS, we set the screening parameter $\tau_t$ as $(\tau_\infty^2 + (1 - \alpha/4)^{t-1} 100^2)^{0.5}$, where $\tau_\infty$ was chosen for each model and dataset, but was fixed for each Byzantine fraction $\delta$ and each random trial. The details of the settings of $\tau_\infty$ can be found in Section D.2.

**Implemented attack algorithms.** We implemented standard five different types of attacks: Bit Flipping Attack (BF), Label Flipping Attack (LF), Mimic Attack (mimic) (Karimireddy et al., 2022), Inner Product Manipulation (IPM) Attack (Xie et al., 2020), and A Little Is Enough (ALIE) Attack (Baruch et al., 2019). The details of these algorithms are found in Section D.3.

**Evaluation.** We compare the test accuracy of the implemented aggregation algorithms. The results of other metrics (train loss, train accuracy, and test loss) can be found in Section D.4. We ran five experimental trials independently and report their 95% confidence intervals.

**Experiment 1: Comparison of the six aggregations for a fixed Byzantine fraction.** Here, we compare the test accuracy of the six aggregations against the five attacks for the case of $\delta = 3/20$. Table 1 shows the best test accuracy comparison of the six aggregations against five attacks. From the "Worst" column, MS outperformed the other methods, including the state-of-the-art CClip and the standard robust aggregations with bucketing, in terms of the *worst* best test accuracy against the five attacks for each model and dataset. This strongly suggests the remarkable robustness of MS. Note that the performances of Avg were severely degraded against ALIE, although Avg worked well against the other attacks, probably because of the relatively small size of the Byzantine fraction $\delta = 3/20$. In the experiments, ALIE seemed to be the strongest attacker, but MS was still robust against it.

**Experiment 2: Comparison of MS with CClip for varied Byzantine fractions.** To further investigate the superiority of our proposed method over the state-of-the-art existing algorithm CClip,

---

[8]Our decision is supported by the empirical effectivity of momentum reported in Karimireddy et al. (2022).

[9]This is because it was reported in Karimireddy et al. (2022) that CClip performed similarly with and without bucketing (probably because CClip without bucketing was already minimax optimal in their settings), and MS with bucketing is not analyzed theoretically, although the analysis will be similar.

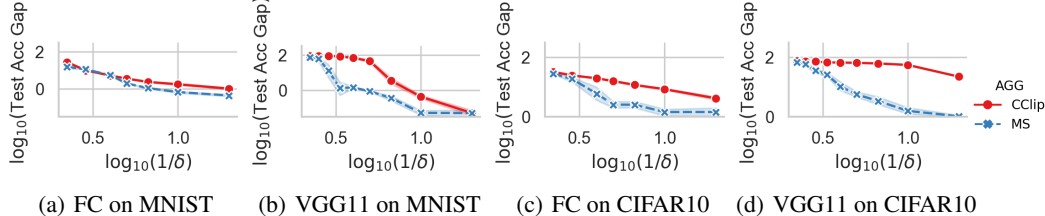

(a) FC on MNIST   (b) VGG11 on MNIST   (c) FC on CIFAR10   (d) VGG11 on CIFAR10

Figure 1: Comparison of the worst relative test accuracy to AVG without Byzantine workers of CClip and MS against the Byzantine fractions $\delta \in \{1/20, 2/20, 3/20, 4/20, 5/20, 7/20, 9/20\}$ for five attacks (BF, LF, mimic, IPM, and ALIE) for each model and dataset (lower is better). The $x$-axis shows $1/\delta$ (i.e., the further to the right on the x-axis, the smaller $\delta$), and both axes are plotted on logarithmic scales.

which is a natural comparator of our method, in particular for relatively small $\delta$, we compare the worst value of the best test accuracy against the five attacks for each Byzantine fraction $\delta \in \{1/20, 2/20, 3/20, 4/20, 5/20, 7/20, 9/20\}$. Figure 1 compares the worst value of the best test accuracy relative to the best test accuracy of Avg aggregation without Byzantine workers, i.e., the difference between the ideal best test accuracy and the worst best test accuracy against five attacks, for each Byzantine fraction. We plotted both axes as logarithmic scales. The results show that MS consistently outperformed CClip by a large margin, particularly for small $\delta$. This matches our theoretical findings because the optimization error of CClip has a factor of $\delta$ but that of MS has a factor of $\delta^2$. We can also see that our method with the fixed clipping radius scheduling performed well for different $\delta$ on each model and dataset thanks to its adaptivity to $\delta$ (Remark 3).

In summary, we conclude that MS outperformed the existing robust algorithms in our experimental settings and the numerical results support our theoretical findings.

## 7 LIMITATIONS AND FUTURE WORK

Here, we briefly discuss some limitations and future directions of this study. First, from a theoretical point of view, our method has no minimax optimal guarantees for $\mathcal{C}_{\mathrm{MH}}(\zeta_{\mathrm{mean}})$, although the minimax optimality for $\mathcal{C}_{\mathrm{UH}}(\zeta_{\max})$ has been shown. Thus, if $\zeta_{\max}/\zeta_{\mathrm{mean}} \gg 1$ (e.g., due to the presence of an outlier worker) or $\delta = \Omega(1)$, our method may perform worse than CClip. In general, which of MS and CClip performs better depends on the type of heterogeneity of the local datasets and Byzantine fraction $\delta$. It is important to construct an algorithm that is minimax optimal for both $\mathcal{C}_{\mathrm{UH}}(\zeta_{\max})$ and $\mathcal{C}_{\mathrm{MH}}(\zeta_{\mathrm{mean}})$ without naively combining MS with CClip or standard aggregations with bucketing. Second, MS requires a communication per model update, which imposes a large communication cost. Therefore, for practical federated learning, our algorithm needs to be extended to allow multiple local updates to reduce the communication cost. In general, multiple local updates may increase the aggregation error because the outputs of honest workers may be very different compared with the case of a single local update when the local datasets are heterogeneous. Thus, it is important to investigate the effect of multiple local updates and the possibility of reducing the communication cost without harming the best optimization error $O(\delta^2 \zeta_{\max}^2)$ of the single local update case.

## 8 CONCLUSION

In this study, we have considered nonconvex federated learning problems with the existence of Byzantine workers. We have proposed a new simple Byzantine robust algorithm called Momentum Screening (MS), which is adaptive to Byzantine fraction $\delta < 0.5$, and shown that it achieves the best optimization error of $O(\delta^2 \zeta_{\max}^2)$ for function class $\mathcal{C}_{\mathrm{UH}}(\zeta_{\max})$, which can be better than the best known error rate of $O(\delta \zeta_{\mathrm{mean}}^2)$ for function class $\mathcal{C}_{\mathrm{MH}}(\zeta_{\mathrm{mean}})$ when Byzantine fraction $\delta$ satisfies $\delta \leq (\zeta_{\mathrm{mean}}/\zeta_{\max})^2$. Furthermore, we have also derived a lower bound for $\mathcal{C}_{\mathrm{UH}}(\zeta_{\max})$ and shown the minimax optimality of our proposed method for $\mathcal{C}_{\mathrm{UH}}(\zeta_{\max})$. In numerical experiments, MS outperformed the existing robust aggregations and the results verified our theoretical findings. Finally, we discussed some limitations of this study and future directions.

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

# A    Convergence Analysis

In this section, convergence analysis of Algorithm 1 is given. $\widetilde{O}$ and $\widetilde{\Theta}$ abbreviate extra poly-logarithmic factors depending on $d$, $1/q$, $T$, $\log|\mathcal{G}|$, $\log(1/\sigma^2)$, $\log G$, $\log(1/\alpha)$, where $q \in (0, 1)$ is the confidence parameter of high probability bound. From the definition of $\alpha$ and $\eta$ in Theorem 1, we can see that $\log(1/\alpha) = O(\log T)$.

**Lemma 2** (Azume-Hoeffding's inequality for norm-subGaussian (Jin et al., 2019)). *Let $X_1, \ldots, X_n$ be random vectors in $\mathbb{R}^d$. Suppose that $\{X_i\}_{i=1}^n$ satisfies the following conditions:*

$$\mathbb{E}[X_i] = 0 \text{ and } \mathbb{P}(\|X_i\| \geq s) \leq 2e^{-\frac{s^2}{2\sigma_i^2}}, \forall s \in \mathbb{R}, \forall i \in [n]$$

*for $\{\sigma_i\}_{i=1}^n$ Then, for any $q \in (0, 1)$, with probability at least $1 - q$ it holds that*

$$\left\|\sum_{i=1}^n X_i\right\| \leq c\sqrt{\sum_{i=1}^n \sigma_i^2 \log\frac{2d}{q}}$$

*for some universal constant $c > 0$.*

**Lemma 3** (Martingle version of Azume-Hoeffding's inequality for norm-subGaussian (Jin et al., 2019)). *Let $X_1, \ldots, X_n$ be random vectors in $\mathbb{R}^d$. Suppose that $\{X_i\}_{i=1}^n$ and corresponding filtrations $\{\mathfrak{F}_i\}_{i=1}^n$ satisfy the following conditions:*

$$\mathbb{E}[X_i \mid \mathfrak{F}_{i-1}] = 0 \text{ and } \mathbb{P}(\|X_i\| \geq s \mid \mathfrak{F}_{i-1}) \leq 2e^{-\frac{s^2}{2\sigma_i^2}}, \forall s \in \mathbb{R}, \forall i \in [n]$$

*for random variables $\{\sigma_i\}_{i=1}^n$ with $\sigma_i \in \mathfrak{F}_{i-1}(i \in [n])$. Then, for any $q \in (0, 1)$ and $A > a > 0$, with probability at least $1 - q$ it holds that*

$$\sum_{i=1}^n \sigma_i^2 \geq A \text{ or } \left\|\sum_{i=1}^n X_i\right\| \leq c\sqrt{\max\left\{\sum_{i=1}^n \sigma_i^2, a\right\}\left(\log\frac{2d}{q} + \log\log\frac{A}{a}\right)}$$

*for some universal constant $c > 0$.*

## A.1    Proof of Lemma 1

Let $t \geq 1$. From $L$-smoothness of $f$, we have

$$f(x^t) \leq f(x^{t-1}) + \langle \nabla f(x^{t-1}), x^t - x^{t-1}\rangle + \frac{L}{2}\|x^t - x^{t-1}\|^2$$

$$= f(x^{t-1}) - \eta\langle \nabla f(x^{t-1}), m^t\rangle + \frac{\eta^2 L}{2}\|m^t\|^2$$

$$\leq f(x^{t-1}) - \eta\langle \nabla f(x^{t-1}), m^t\rangle + \frac{\eta}{2}\|m^t\|^2$$

$$= f(x^{t-1}) - \frac{\eta}{2}\|\nabla f(x^{t-1})\|^2 + \frac{\eta}{2}\|m^t - \nabla f(x^{t-1})\|^2$$

$$\leq f(x^{t-1}) - \frac{\eta}{2}\|\nabla f(x^{t-1})\|^2 + \eta\|\bar{e}^t\|^2 + \eta\|m^t - \bar{m}^t\|^2.$$

Here, the second inequality used $\eta \leq 1/L$. This finishes the proof of Lemma 1.

**Lemma 4.** *For every $t \geq 1$, it holds that $\|m_i^t\| \leq 2G$ almost surely.*

*Proof.* When $t = 1$, Assumption 4 immediately implies the desired result because $m_i^1 = g_i^1$.

Let $t \geq 2$. Observe that $\|m_i^t\| = \|(1-\alpha)m_i^{t-1} + \alpha g_i^t\| \leq (1-\alpha)\|m_i^{t-1}\| + \alpha G \leq (1-\alpha)^{t-1}\|m_i^1\| + \sum_{t'=2}^t \alpha(1-\alpha)^{t-t'}G \leq (1-\alpha)^{t-1}G + G \leq 2G$. $\qquad\square$

## A.2 PROOF OF PROPOSITION 1

First observe that $\|\bar{e}^1\|^2 = \|\bar{g}^1 - \nabla f(x^0)\|^2$.

Let $t \geq 2$. Observe that

$$
\begin{aligned}
\|\bar{e}^t\|^2 &= \|\bar{m}^t - \nabla f(x^{t-1})\|^2 \\
&= \|(1-\alpha)\bar{m}^{t-1} + \alpha\bar{g}^t - \nabla f(x^{t-1})\|^2 \\
&= (1-\alpha)^2 \|\bar{m}^{t-1} - \nabla f(x^{t-1})\|^2 + 2\alpha(1-\alpha)\langle\bar{m}^{t-1}, \bar{g}^t - \nabla f(x^{t-1})\rangle \\
&\quad + \alpha^2 \|\bar{g}^t - \nabla f(x^{t-1})\|^2 \\
&\leq (1-\alpha)\|\bar{e}^{t-1}\|^2 + \left(1 + \frac{1}{\alpha}\right)\|\nabla f(x^{t-1}) - \nabla f(x^{t-2})\|^2 \\
&\quad + 2\alpha(1-\alpha)\langle\bar{m}^{t-1}, \bar{g}^t - \nabla f(x^{t-1})\rangle + \alpha^2 \|\bar{g}^t - \nabla f(x^{t-1})\|^2 \\
&\leq (1-\alpha)\|\bar{e}^{t-1}\|^2 + \eta^2 L^2 \left(1 + \frac{1}{\alpha}\right)\|m^{t-1}\|^2 \\
&\quad + 2\alpha(1-\alpha)\langle\bar{m}^{t-1}, \bar{g}^t - \nabla f(x^{t-1})\rangle + \alpha^2 \|\bar{g}^t - \nabla f(x^{t-1})\|^2 \\
&\leq \left(1 - \alpha + 3\eta^2 L^2\left(1 + \frac{1}{\alpha}\right)\right)\|\bar{e}^{t-1}\|^2 + 3\eta^2 L^2 \left(1 + \frac{1}{\alpha}\right)\|m^{t-1} - \bar{m}^{t-1}\|^2 \\
&\quad + 3\eta^2 L^2 \left(1 + \frac{1}{\alpha}\right)\|\nabla f(x^{t-2})\|^2 + 2\alpha(1-\alpha)\langle\bar{m}^{t-1}, \bar{g}^t - \nabla f(x^{t-1})\rangle \\
&\quad + \alpha^2 \|\bar{g}^t - \nabla f(x^{t-1})\|^2.
\end{aligned}
$$

Here, for the first inequality, we used Young's inequality. The second inequality holds from $L$-smoothness of $f$.

Now, we set $\eta$ to be $3\eta^2 L^2(1 + 1/\alpha) \leq \alpha/16$ ($\leq \alpha/2$). In this case, note that $\eta = O(\alpha/L)$. Then, we have

$$
\begin{aligned}
\|\bar{e}^t\|^2 &= \|\bar{m}^t - \nabla f(x^{t-1})\|^2 \\
&\leq \left(1 - \frac{\alpha}{2}\right)\|\bar{e}^{t-1}\|^2 + 3\eta^2 L^2 \left(1 + \frac{1}{\alpha}\right)\|m^{t-1} - \bar{m}^{t-1}\|^2 \\
&\quad + 3\eta^2 L^2 \left(1 + \frac{1}{\alpha}\right)\|\nabla f(x^{t-2})\|^2 + 2\alpha(1-\alpha)\langle\bar{m}^{t-1}, \bar{g}^t - \nabla f(x^{t-1})\rangle \\
&\quad + \alpha^2 \|\bar{g}^t - \nabla f(x^{t-1})\|^2 \\
&= \left(1 - \frac{\alpha}{4}\right)\|\bar{e}^{t-1}\|^2 - \frac{\alpha}{4}\|\bar{e}^{t-1}\|^2 + 3\eta^2 L^2 \left(1 + \frac{1}{\alpha}\right)\|m^{t-1} - \bar{m}^{t-1}\|^2 \\
&\quad + 3\eta^2 L^2 \left(1 + \frac{1}{\alpha}\right)\|\nabla f(x^{t-2})\|^2 + 2\alpha(1-\alpha)\langle\bar{m}^{t-1}, \bar{g}^t - \nabla f(x^{t-1})\rangle \\
&\quad + \alpha^2 \|\bar{g}^t - \nabla f(x^{t-1})\|^2.
\end{aligned}
$$

Recursively using this inequality, we obtain

$$
\|\bar{e}^t\|^2 = \|\bar{m}^t - \nabla f(x^{t-1})\|^2
$$

$$
\leq \left(1 - \frac{\alpha}{4}\right)^{t-1} \|\bar{e}^1\|^2 - \frac{\alpha}{4} \sum_{\tau=2}^{t} \left(1 - \frac{\alpha}{4}\right)^{t-\tau} \|\bar{e}^{\tau-1}\|^2
$$

$$
+ 3\eta^2 L^2 \left(1 + \frac{1}{\alpha}\right) \sum_{\tau=2}^{t} \left(1 - \frac{\alpha}{4}\right)^{t-\tau} \|m^{\tau-1} - \bar{m}^{\tau-1}\|^2
$$

$$
+ 3\eta^2 L^2 \left(1 + \frac{1}{\alpha}\right) \sum_{\tau=2}^{t} \left(1 - \frac{\alpha}{4}\right)^{t-\tau} \|\nabla f(x^{\tau-2})\|^2 \tag{1}
$$

$$
+ 2\alpha(1 - \alpha) \sum_{\tau=2}^{t} \left(1 - \frac{\alpha}{4}\right)^{t-\tau} \langle \bar{m}^{\tau-1}, \bar{g}^\tau - \nabla f(x^{\tau-1}) \rangle
$$

$$
+ \sum_{\tau=2}^{t} \left(1 - \frac{\alpha}{4}\right)^{t-\tau} \alpha^2 \|\bar{g}^\tau - \nabla f(x^{\tau-1})\|^2.
$$

To further bound the right hand of (1), we apply concentration inequalities.

**Bounding the first and the last term of the r.h.s. of (1)** Since $g_i^t$ is independent norm sub-Gaussian with parameter $\sigma^2$ for every $i \in \mathcal{G}$, applying Lemma 2 gives

$$
\|\bar{g}^t - \nabla f(x^{t-1})\|^2 \leq \frac{c^2 \sigma^2 \log \frac{2d}{q}}{|\mathcal{G}|}
$$

with probability at least $1 - q$. Using union bound for $t \in [T]$, we obtain

$$
\forall t \in [T] : \|\bar{g}^t - \nabla f(x^{t-1})\|^2 \leq \frac{c^2 \sigma^2 \log \frac{2Td}{q}}{|\mathcal{G}|}
$$

with probability at least $1 - q$.

**Bounding the 5th term of the r.h.s. of (1)** Let $X_\tau^i := \left(1 - \frac{\alpha}{4}\right)^{t-\tau} \langle \bar{m}^{\tau-1}, g_i^\tau - \nabla f_i(x^{\tau-1}) \rangle$ and $\bar{X}_\tau := (1/|\mathcal{G}|) \sum_{i \in \mathcal{G}} X_\tau^i$. We apply Lemma 3 to $\{X_\tau^i\}_{\tau \in [T], i \in \mathcal{G}}$. To do so, we check the assumptions of Lemma 3. At first, it is easy to see that $\mathbb{E}[X_\tau^i | \mathcal{F}_{\tau-1}] = 0$ because $\mathbb{E}[g_i^\tau | \mathcal{F}_{\tau-1}] = \nabla f_i(x^{\tau-1})$. Also, from the norm sub-Gaussian properties of $g_i^\tau$, we have

$$
\mathbb{P}(X_\tau^i \geq s | \mathcal{F}_{\tau-1}) \leq \mathbb{P}\left(\|g_i^\tau - \nabla f_i(x^{\tau-1})\| \geq \frac{s}{(1 - \alpha/4)^{t-\tau} \|\bar{m}^{\tau-1}\|} \Big| \mathcal{F}_{\tau-1}\right)
$$

$$
\leq 2 \exp\left(-\frac{s^2}{2\sigma^2 (1 - \alpha/4)^{2(t-\tau)} \|\bar{m}^{\tau-1}\|^2}\right).
$$

Thus, the assumptions of Lemma 3 are satisfied with $\sigma_\tau \leftarrow \sigma(1 - \alpha/4)^{t-\tau} \|\bar{m}^{\tau-1}\|$.

Observe that $|\mathcal{G}| \sum_{\tau=2}^{t} \sigma_\tau^2 \leq 16|\mathcal{G}|\sigma^2 G^2/\alpha$ almost surely from Lemma 4. Apply Lemma 3 with $A \leftarrow 32|\mathcal{G}|\sigma^2 G^2/\alpha$ and $a \leftarrow \varepsilon$ yields

$$
\left|\sum_{\tau=2}^{t} \bar{X}_\tau\right| \leq \frac{c}{\sqrt{|\mathcal{G}|}} \sqrt{\left(\sum_{\tau=2}^{t} \sigma_\tau^2 + \varepsilon\right)\left(\log \frac{2d}{q} + \log\log \frac{A}{\varepsilon}\right)}
$$

$$
\leq \frac{c}{\sqrt{|\mathcal{G}|}} \sqrt{\sigma^2 \sum_{\tau=2}^{t} (1 - \alpha/4)^{2(t-\tau)} \|\bar{m}^{\tau-1}\|^2 \left(\log \frac{2d}{q} + \log\log \frac{A}{\varepsilon}\right) + \widetilde{O}\left(\sqrt{\varepsilon}\right)}
$$

$$
\leq \frac{c\sigma^2}{\gamma |\mathcal{G}|} \left(\log \frac{2d}{q} + \log\log \frac{A}{\varepsilon}\right) + c\gamma \sum_{\tau=2}^{t} (1 - \alpha/4)^{t-\tau} \|\bar{m}^{\tau-1}\|^2 + \widetilde{O}\left(\sqrt{\varepsilon}\right)
$$

$$
\leq \frac{c\sigma^2}{\gamma |\mathcal{G}|} \left(\log \frac{2d}{q} + \log\log \frac{A}{\varepsilon}\right) + 2c\gamma \sum_{\tau=2}^{t} (1 - \alpha/4)^{t-\tau} (\|\bar{e}^{\tau-1}\|^2 + \|\nabla f(x^{\tau-1})\|^2) + \widetilde{O}\left(\sqrt{\varepsilon}\right).
$$

Here, for the third inequality, we used $\sqrt{ab} \leq \sqrt{a^2/(2\gamma^2) + \gamma^2 b^2/2} \leq a/\gamma + \gamma b$ for any $\gamma > 0$. We choose $\gamma$ such that $4\alpha(1-\alpha)c\gamma \leq \alpha/4$ to cancel out the second term of the r.h.s. of $|\sum_{\tau=2}^{t} \bar{X}_\tau|$ by the second term of the r.h.s. of (1).

Combining these bounds, we obtain

$$\|\bar{e}^t\|^2 \leq O\left(\frac{\eta^2 L^2}{\alpha}\right) \sum_{\tau=2}^{t} \left(1 - \frac{\alpha}{4}\right)^{t-\tau} \|m^{\tau-1} - \bar{m}^{\tau-1}\|^2$$

$$+ \alpha\left(\frac{1}{16} + 4c\gamma(1-\alpha)\right) \sum_{\tau=2}^{t} \left(1 - \frac{\alpha}{4}\right)^{t-\tau} \|\nabla f(x^{\tau-2})\|^2$$

$$+ \widetilde{O}\left(\left(\left(1 - \frac{\alpha}{4}\right)^{t-1} + \frac{\alpha}{\gamma}\right) \frac{\sigma^2}{|\mathcal{G}|}\right) + \widetilde{O}\left(\sqrt{\varepsilon}\right).$$

Finally, taking $\varepsilon \leftarrow (\alpha^2/\gamma^2)(\sigma^2/|\mathcal{G}|)$ and substituting $\gamma := 1/(128c(1-\alpha))$ finishes the proof of Proposition 1.

### A.3 PROOF OF PROPOSITION 2

In the following proof, we fix $t \in [T]$ and omit index $t$ as long as it does not cause confusion for simple presentation.

First we show that $\mathcal{G} \subset \hat{\mathcal{G}}$ for Algorithm 2. To do so, we fix any $i \in \mathcal{G}$ and show that $i \in \hat{\mathcal{G}}$. First note that $\mathcal{G} \subset \{j \in [n] : \|m_i - m_j\| \leq \rho_{\max}\} \subset \{j \in [n] : \|m_i - m_j\| \leq \tau\}$ since $\tau \geq \rho_{\max}$. Then, since $\delta < 0.5$, we know that $|\mathcal{G}| \geq n/2$. Thus, $|\{j \in [n] : \|m_i - m_j\| \leq \tau\}| \geq n/2$. Hence, from the definition of $\hat{\mathcal{G}}$, $i \in \hat{\mathcal{G}}$.

Next, we show that $\mathcal{G} \subset \hat{\mathcal{G}}$ for Algorithm 3. Note that Algorithm 4 always terminates in a finite number of iterations because $\tau \geq \rho_{\max}$ is assumed. Let $i \in \mathcal{G}$ be fixed. Let $p := \mathbb{P}(j_{\text{ref}}^k \in \mathcal{B})$. Observe that random variable $X_k := \mathbb{I}_{j_{\text{ref}}^k \in \mathcal{B}}$ independently follows Bernoulli distribution with parameter $p$ for $k \in K$. Then, from Hoeffding's inequality, we have $|\sum_{k=1}^{K}(X_k - p)| \leq \sqrt{(K/2)\log(2/\delta)}$ with probability at least $1 - q$. Observe that

$$\|m_i - m_{\text{ref}}\| \leq \left\|\frac{1}{|\mathcal{J}|} \sum_{k:j_{\text{ref}}^k \in \mathcal{J} \cap \mathcal{G}} (m_i - m_{j_{\text{ref}}^k})\right\| + \left\|\frac{1}{|\mathcal{J}|} \sum_{k:j_{\text{ref}}^k \in \mathcal{J} \cap \mathcal{B}} (m_i - m_{j_{\text{ref}}^k})\right\|$$

$$\leq \frac{|\mathcal{J} \cap \mathcal{G}|\rho_{\max} + |\mathcal{J} \cap \mathcal{B}|(\rho_{\max} + \tau)}{K}$$

$$\leq \frac{|\mathcal{J} \cap \mathcal{G}| + 2|\mathcal{J} \cap \mathcal{B}|}{K}\tau.$$

Here, for the second inequalities, we used the fact that for $j_{\text{ref}}^k \in \mathcal{B}$, $\|m_i - m_{j_{\text{ref}}^k}\| \leq \|m_i - m_{j'}\| + \|m_{j'} - m_{j_{\text{ref}}}\| \leq \rho_{\max} + \tau$, where $j' \in \mathcal{G}$ is some worker index such that $\|m_{j'} - m_{j_{\text{ref}}}\| \leq \tau$ and its existence can be guaranteed from the procedures of Algorithm 4. Then, from the high probability bound, we have $|\mathcal{J} \cap \mathcal{G}| \leq (1-p)K + \sqrt{(K/2)\log(2/\delta)}$ and $|\mathcal{J} \cap \mathcal{B}| \leq pK + \sqrt{(K/2)\log(2/\delta)}$. Using these results and $p \leq \delta$, we obtain

$$\|m_i - m_{\text{ref}}\| \leq \left(1 + \delta + 3\sqrt{\frac{\log(2/\delta)}{2K}}\right)\tau \leq (1+s)\tau$$

with probability at least $1 - \delta$ when

$$s \geq \delta + 3\sqrt{\frac{\log(2/\delta)}{2K}}.$$

Hence, for both Algorithms 2 and 3, we have $i \in \hat{\mathcal{G}}$ and thus $\mathcal{G} \subset \hat{\mathcal{G}}$.

Now, we analyze the aggregation error. Let $\mathcal{B}$ be the set of all the Byzantine workers.

$$\|m - \bar{m}\|^2 = \left\| \frac{1}{|\hat{\mathcal{G}}|} \sum_{i \in \hat{\mathcal{G}}} m_i - \frac{1}{|\mathcal{G}|} \sum_{i \in \mathcal{G}} m_i \right\|^2$$

$$= \left\| \frac{1}{|\hat{\mathcal{G}}|} \sum_{i \in \hat{\mathcal{G}}} m_i - (1 - \frac{|\mathcal{G}|}{|\hat{\mathcal{G}}|})\bar{m} - \frac{1}{|\hat{\mathcal{G}}|} \sum_{i \in \mathcal{G}} m_i \right\|^2$$

$$= \left\| \frac{1}{|\hat{\mathcal{G}}|} \sum_{i \in \hat{\mathcal{G}}} m_i - \frac{|\hat{\mathcal{G}} \cap \mathcal{B}|}{|\hat{\mathcal{G}}|}\bar{m} - \frac{1}{|\hat{\mathcal{G}}|} \sum_{i \in \mathcal{G}} m_i \right\|^2$$

$$= \left\| \frac{1}{|\hat{\mathcal{G}}|} \sum_{i \in \hat{\mathcal{G}} \cap \mathcal{B}} (m_i - \bar{m}) \right\|^2$$

$$\leq \left( \frac{1}{|\hat{\mathcal{G}}|} \sum_{i \in \hat{\mathcal{G}} \cap \mathcal{B}} \|m_i - \bar{m}\| \right)^2 .$$

Here, for the third and fourth equality, we used the fact that $\hat{\mathcal{G}} \setminus \mathcal{G} = \hat{\mathcal{G}} \cap \mathcal{B}$ and thus $|\hat{\mathcal{G}}| - |\mathcal{G}| = |\hat{\mathcal{G}} \setminus \mathcal{G}| = |\hat{\mathcal{G}} \cap \mathcal{B}|$.

Thus, for Algorithm 2, we have

$$\|m - \bar{m}\|^2 \leq \left( \frac{1}{|\hat{\mathcal{G}}|} \sum_{i \in \hat{\mathcal{G}} \cap \mathcal{B}} (\|m_i - m_{j_i}\| + \|m_{j_i} - \bar{m}\|) \right)^2$$

$$\leq 4\delta^2 \tau^2.$$

Here, in the first inequality, $j_i$ is some index satisfying $j_i \in \mathcal{G}$ and $\|m_i - m_{j_i}\| \leq \tau$. Note that for each $i \in \hat{\mathcal{G}}$, such $j_i$ always exists because $|\{j \in [n] : \|m_i - m_j\| \leq \tau\}| \geq 0.5n$ and $|\mathcal{B}| < 0.5n$. The last inequality holds from $\|m_i - m_{j_i}\| \leq \tau$ and $\|m_{j_i} - \bar{m}\| = (1/|\mathcal{G}|) \sum_{k \in \mathcal{G}} \|m_{j_i} - m_k\| \leq \rho_{\max} \leq \tau$ and $|\hat{\mathcal{G}} \cap \mathcal{B}|/|\hat{\mathcal{G}}| = 1 - |\mathcal{G}|/|\hat{\mathcal{G}}| \leq 1 - |\mathcal{G}|/n \leq \delta$.

Similarly, for Algorithm 3, we have

$$\|m - \bar{m}\|^2 \leq \left( \frac{1}{|\hat{\mathcal{G}}|} \sum_{i \in \hat{\mathcal{G}} \cap \mathcal{B}} (\|m_i - m_{\mathrm{ref}}\| + \|m_{\mathrm{ref}} - \bar{m}\|) \right)^2$$

$$\leq 4 (1 + s)^2 \delta^2 \tau^2$$

with probability at least $1 - q$.

Here, in the last inequality, we used $\|m_i - m_{\mathrm{ref}}\| \leq (1 + s)\tau$ for $i \in \hat{\mathcal{G}}$ (note that $\mathcal{G} \subset \mathcal{G}$), and $|\hat{\mathcal{G}} \cap \mathcal{B}|/|\hat{\mathcal{G}}| = 1 - |\mathcal{G}|/|\hat{\mathcal{G}}| \leq 1 - |\mathcal{G}|/n \leq \delta$.

This completes all the proofs of Proposition 2.

### A.4 PROOF OF PROPOSITION 3

Observe that
$$\|m_i^t - m_j^t\|^2 = \|(1 - \alpha)(m_i^{t-1} - m_j^{t-1}) + \alpha(g_i^t - g_j^t)\|^2$$

$$= (1 - \alpha)^2 \|m_i^{t-1} - m_j^{t-1}\|^2 + 2\alpha(1 - \alpha)\langle m_i^{t-1} - m_j^{t-1}, g_i^t - g_j^t \rangle + \alpha^2 \|g_i^t - g_j^t\|^2$$

$$\leq ((1 - \alpha)^2 + \gamma'\alpha(1 - \alpha))\|m_i^{t-1} - m_j^{t-1}\|^2 + \left( \frac{\alpha(1 - \alpha)}{\gamma'} + 2\alpha^2 \right) \zeta_{\max}^2$$

$$+ 4\alpha^2(\|g_i^t - \nabla f_i(x_{t-1})\|^2 + \|g_j^t - \nabla f_j(x_{t-1})\|^2)$$

$$+ 2\alpha(1 - \alpha)\langle m_i^{t-1} - m_j^{t-1}, g_i^t - \nabla f_i(x^{t-1}) + g_j^t - \nabla f_j(x^{t-1}) \rangle.$$

Here, we set $\gamma' := 1/2$. Then, the coefficient of the first term is bounded by $1 - \alpha/2$. We decompose this as $(1 - \alpha/4) - \alpha/4$ and then recursively use the inequality:

$$
\begin{aligned}
\|m_i^t - m_j^t\|^2 &\leq ((1-\alpha)^2 + \gamma\alpha(1-\alpha))\|m_i^{t-1} - m_j^{t-1}\|^2 + \left(2\alpha(1-\alpha) + 2\alpha^2\right)\zeta_{\max}^2 \\
&\quad + 2\alpha(1-\alpha)\langle m_i^{t-1} - m_j^{t-1}, g_i^t - \nabla f_i(x^{t-1}) + g_j^t - \nabla f_j(x^{t-1})\rangle \\
&\quad + 4\alpha^2(\|g_i^t - \nabla f_i(x_{t-1})\|^2 + \|g_j^t - \nabla f_j(x_{t-1})\|^2) \\
&\leq (1-\alpha/4)\|m_i^{t-1} - m_j^{t-1}\|^2 - \frac{\alpha}{4}\|m_i^{t-1} - m_j^{t-1}\|^2 + \left(2\alpha(1-\alpha) + 2\alpha^2\right)\zeta_{\max}^2 + 4\alpha^2\sigma^2 \\
&\quad + 2\alpha(1-\alpha)\langle m_i^{t-1} - m_j^{t-1}, g_i^t - \nabla f_i(x^{t-1}) + g_j^t - \nabla f_j(x^{t-1})\rangle \\
&\quad + 4\alpha^2(\|g_i^t - \nabla f_i(x_{t-1})\|^2 + \|g_j^t - \nabla f_j(x_{t-1})\|^2) \\
&\leq (1-\alpha/4)^{t-1}\|m_i^1 - m_j^1\|^2 - \frac{\alpha}{4}\sum_{\tau=2}^t (1-\alpha/4)^{t-\tau}\|m_i^{\tau-1} - m_j^{\tau-1}\|^2 + 12\zeta_{\max}^2 \\
&\quad + 2\alpha(1-\alpha)\sum_{\tau=2}^t (1-\alpha/4)^{t-\tau}\langle m_i^{\tau-1} - m_j^{\tau-1}, g_i^\tau - \nabla f_i(x^{\tau-1}) + g_j^\tau - \nabla f_j(x^{\tau-1})\rangle \\
&\quad + 4\alpha^2\sum_{\tau=2}^t (1-\alpha/4)^{t-\tau}(\|g_i^\tau - \nabla f_i(x_{\tau-1})\|^2 + \|g_j^\tau - \nabla f_j(x_{\tau-1})\|^2).
\end{aligned}
$$

$$(2)$$

To bound the first term, last term, and the fourth term, we again apply Lemma 2 and Lemma 3. Similar to the arguments of the proof of Lemma 1, we have

$$
\forall t \in [T] : \|g_i^t - \nabla f(x^{t-1})\|^2 \leq c^2\sigma^2 \log\frac{2Td}{q}
$$

with probability at least $1 - q$. Also, defining $X_\tau^{i,j} := \left(1 - \frac{\alpha}{4}\right)^{t-\tau}\langle m_i^{\tau-1} - m_j^{\tau-1}, g_i^\tau - \nabla f_i(x^{\tau-1})\rangle$, with $A \leftarrow 64\sigma^2 G^2/\alpha$ and $a \leftarrow \varepsilon$ we have

$$
\left|\sum_{\tau=2}^t X_\tau^{i,j}\right| \leq \frac{c\sigma^2}{\gamma}\left(\log\frac{2d}{q} + \log\log\frac{A}{\varepsilon}\right) + c\gamma\sum_{\tau=2}^t (1-\alpha/4)^{t-\tau}\|m_i^{\tau-1} - m_j^{\tau-1}\|^2 + \widetilde{O}\left(\sqrt{\varepsilon}\right),
$$

where $\gamma'' > 0$ is arbitrary. Similar arguments for $\check{X}_\tau^{i,j} := \left(1 - \frac{\alpha}{4}\right)^{t-\tau}\langle m_i^{\tau-1} - m_j^{\tau-1}, g_j^\tau - \nabla f_j(x^{\tau-1})\rangle$ gives the same bound as $|\sum_{\tau=2}^t \check{X}_\tau^{i,j}|$. We choose $\gamma'' = \Theta(1)$ such that $4\alpha(1-\alpha)c\gamma'' \leq \alpha/4$ to cancel out the second terms of the r.h.s. of $|\sum_{\tau=2}^t X_\tau^{i,j}|$ and $|\sum_{\tau=2}^t \check{X}_\tau^{i,j}|$ by the second term the r.h.s. of (2). Note that such $\gamma''$ can be $\Theta(1)$ since we assume $\alpha \leq 1/2$. Finally, we take union bounds for $i, j \in \mathcal{G}$ and set $\varepsilon \leftarrow \alpha^2\sigma^4$. Then, we obtain the desired result.

## A.5 Proof of Theorem 1

First, observe that $\eta \leq \sqrt{\alpha/(48(1 + 1/\alpha))}/L = \Theta(\alpha/L)$ from the definitions of $\eta$ and $\alpha$.

Combining Proposition 2 with Proposition 3 results in

$$
\|m^t - \bar{m}^t\|^2 \leq O(\delta^2\zeta_{\max}^2) + \widetilde{O}(((1 - \Omega(\alpha))^{t-1} + \alpha)\delta^2\sigma^2)
$$

and thus

$$
\sum_{t=1}^T \|m^t - \bar{m}^t\|^2 \leq O\left(T\delta^2\zeta_{\max}^2 + \widetilde{O}\left(\frac{1}{\alpha} + \alpha T\right)\delta^2\sigma^2\right)
$$

with probability at least $1 - 3q$.

From Lemma 1, we have

$$
0 \leq f(x^0) - f(x^T) - \frac{\eta}{2}\sum_{t=1}^T \|\nabla f(x^{t-1})\|^2 + \eta\sum_{t=1}^T \|\bar{e}^t\|^2 + O(\eta T\delta^2\zeta_{\max}^2) + \widetilde{O}\left(\eta\left(\frac{1}{\alpha} + \frac{\alpha T}{\gamma}\right)\delta^2\sigma^2\right).
$$

Then, from Proposition 1, with probability at least $1 - 2q$ we have

$$\sum_{t=1}^{T} \|\bar{e}^t\|^2 \le O\left(\frac{\eta^2 L^2}{\alpha^2}\right) \sum_{t=1}^{T} \|m^{t-1} - \bar{m}^{t-1}\|^2$$

$$+ \left(\frac{1}{4} + 16c\gamma(1-\alpha)\right) \sum_{t=1}^{T} \|\nabla f(x^{t-1})\|^2$$

$$+ \tilde{O}\left(\left(\frac{1}{\alpha} + \frac{\alpha T}{\gamma}\right) \frac{\sigma^2}{|\mathcal{G}|}\right).$$

Let $\gamma := 1/(128c(1-\alpha))$. Note that $\gamma = \Omega(1)$ since $\alpha \le 1/2$. Then, since

$$\frac{1}{4} + 16c\gamma(1-\alpha) \le \frac{3}{8},$$

we obtain

$$\frac{1}{T} \sum_{t=1}^{T} \|\nabla f(x^{t-1})\|^2 \le O\left(\frac{f(x^0) - f(x^T)}{\eta T}\right) + O\left(\left(\frac{\eta^2 L^2}{\alpha^2} + 1\right) \delta^2 \zeta_{\max}^2\right)$$

$$+ \tilde{O}\left(\left(\frac{\eta^2 L^2}{\alpha^2} + 1\right) \delta^2 \left(\frac{1}{\alpha T} + \alpha\right) \sigma^2 + \left(\frac{1}{\alpha T} + \alpha\right) \frac{\sigma^2}{n}\right)$$

with probability at least $1 - 5q$. Substituting the definition of $\alpha$ gives the desired results.

## B   LOWER BOUND FOR $\mathcal{C}_{\mathrm{UH}}(\zeta_{\max})$

**Lemma 5.** *Let $x_0 > 0$ and $h : \mathbb{R} \to \mathbb{R}$ be $L$-smooth, and $G$-Lipschitz on $|x| \le x_0$. If we define $\widetilde{h}$ as*

$$\tilde{h}(x) := \begin{cases} h(x) & (|x| \le x_0) \\ h'(x_0)(x - x_0) + h(x_0) & (x > x_0) \\ h'(-x_0)(x + x_0) + h(x_0) & (x < -x_0) \end{cases},$$

*$\widetilde{h}$ is $L$-smooth and $G$-Lipschitz on $\mathbb{R}$.*

*Proof.* It is obvious that $\widetilde{h}$ is $G$-Lipchitzness because $\widetilde{h}$ is linear on $|x| \ge x_0$ and $|h'(x_0)|, |h'(-x_0)| \le G$.

We show $L$-smoothness of $\widetilde{h}$. To show this, we pick any $x, y \in \mathbb{R}$ and show that $|\tilde{h}'(x) - \tilde{h}'(y)| \le L|x - y|$.

If $x, y \in \{x \in \mathbb{R} : |x| \le x_0\}$ or $x, y \in \{x \in \mathbb{R} : x > x_0\}$ or $x, y \in \{x \in [r] : x < -x_0\}$, $L$-smoothness is trivial because of the $L$-smoothness of $h$ and linearity of $\widetilde{h}$ on $|x| \ge x_0$. Without loss of generality, we assume $x < y$. Then, we only need to consider the following three cases:

- $|x| \le x_0$ and $y > x_0$: In this case, we have $|\tilde{h}'(x) - \tilde{h}'(y)| = |h'(x) - h'(x_0)| \le L|x - x_0| \le L|x - y|$.

- $x < -x_0$ and $|y| \le x_0$: In this case, we have $|\tilde{h}'(x) - \tilde{h}'(y)| = |h'(-x_0) - h'(y)| \le L|y + x_0|$. Observe that $|y + x_0| + |x + x_0| = y + x_0 - (x + x_0) = y - x = |x - y|$ and thus $|y + x_0| \le |x - y|$.

- $x < -x_0$ and $y > x_0$: In this case, we have $|\tilde{h}'(x) - \tilde{h}'(y)| = |h'(x_0) - h'(-x_0)| \le 2L|x_0| \le L|x - y|$.

This finishes the proof of Lemma 5. $\qquad\square$

## B.1 PROOF OF THEOREM 2

Let $d = 1$ and $G \geq 2\zeta_{\max}$. We define the following two local objectives $\{f_i^1\}_{i=1}^n$ and $\{f_i^2\}_{i=1}^n$:

$$f_i^1(x) := \begin{cases} \tilde{h}_a(x) & (i \in \{1, \ldots, \delta n\}), \\ \tilde{h}_b(x) & (otherwise) \end{cases}$$

and

$$f_i^2(x) := \begin{cases} \tilde{h}_b(x) & (i \in \{1, \ldots, (1-\delta)n\}), \\ \tilde{h}_a(x) & (otherwise) \end{cases}.$$

Here, $\widetilde{h}_a$ is defined as the smoothed function of $h_a(x) := \frac{L}{2}x^2 - \zeta_{\max}x$ by Lemma 5 with $x_0 := G/(2L)$. Similarly, $\widetilde{h}_b$ is defined as the smoothed function of $h_b(x) := \frac{L}{2}x^2$ by Lemma 5 with $x_0 := G/(2L)$. Note that $h_a$ and $h_b$ is $L$-smooth on $\mathbb{R}$ and $G$-Lipschitz on $|x| \leq x_0$.

Then, the objective functions become

$$f^1(x) = \frac{1}{(1-\delta)n} \sum_{i=1}^{(1-\delta)n} f_i^1(x) = \begin{cases} \frac{L}{2}x^2 - \frac{\delta}{1-\delta}\zeta_{\max}x & (|x| \leq x_0) \\ (Lx_0 - \frac{\delta}{1-\delta}\zeta_{\max})(x - x_0) + \frac{L}{2}x_0^2 - \frac{\delta}{1-\delta}\zeta_{\max}x_0 & (x > x_0) \\ (-Lx_0 - \frac{\delta}{1-\delta}\zeta_{\max})(x + x_0) + \frac{L}{2}x_0^2 + \frac{\delta}{1-\delta}\zeta_{\max}x_0 & (x < -x_0) \end{cases}$$

and

$$f^2(x) = \frac{1}{(1-\delta)n} \sum_{i=1}^{(1-\delta)n} f_i^2(x) = \begin{cases} \frac{L}{2}x^2 & (|x| \leq x_0) \\ Lx_0(x - x_0) + \frac{L}{2}x_0^2 & (x > x_0) \\ -Lx_0(x + x_0) + \frac{L}{2}x_0^2 & (x > x_0) \end{cases}.$$

Moreover, we assume that stochastic gradient $g_i$ of $f_i^k$ matches $f_i^k$ itself, i.e., the deterministic oracle is considered for each $k = 1, 2$.

Now, we check $\{f_i^1\}_{i \in \mathcal{G}}, \{f_i^2\}_{i \in \mathcal{G}} \in \mathcal{C}_{\mathrm{UH}}(\zeta_{\max})$.

- (Assumption 1) $L$-smoothness is immediately satisfied from Lemma 5 since $h_a$ and $b_b$ are $L$-smooth.
- (Assumption 2) Observe that $f^1$ attains minimum $-\delta^2\zeta_{\max}^2/(2L(1-\delta)^2)$ at $x = \delta\zeta_{\max}/((1-\delta)L) \leq x_0$, and $f^2$ attains minimum 0 at $x = 0$.
- (Assumption 3) Since $g_i$ matches $f_i^k$ for each $k = 1, 2$, arbitrary $\sigma^2 > 0$ satisfies the sub-Gaussian property.
- (Assumption 4) We check $G$-Lipschitzness of $h_a$ and $h_b$ on $|x| \leq x_0$. $|h_a'(x)| = |Lx - \zeta_{\max}| \leq L|x| + \zeta_{\max} \leq G/2 + G/2 = G$ for $|x| \leq x_0$ from the definition $x_0 := G/(2L)$ and $G \geq 2\zeta_{\max}$. Similarly, we can see that $|h_b'(x)| = L|x| \leq G/2 \leq G$. Hence, Lemma 5 gives the $G$-Lipschitzness of $\widetilde{h}_a$ and $\widetilde{h}_b$.
- (Assumption 5) First we show that $\{f_i^1\}_{i \in \mathcal{G}}$ satisfies $\zeta_{\max}$-uniform gradient heterogeneity condition. For $i \in \{1, \ldots, \delta n\}$, $|\nabla f_i^1(x) - \nabla f^1(x)| = (1 - 2\delta)/(1 - \delta) \cdot \zeta_{\max} \leq \zeta_{\max}$ for any $x \in \mathbb{R}$. Also, for $i \in \{\delta n+1, \ldots, (1-\delta)n\}$, $|\nabla f_i^1(x) - \nabla f^1(x)| = \delta/(1-\delta) \cdot \zeta_{\max} \leq \zeta_{\max}$ for any $x \in \mathbb{R}$. Thus, $\{f_i^1\}_{i \in \mathcal{G}}$ is $\zeta_{\max}$-uniform gradient heterogeneous. $\zeta_{\max}$-uniform gradient heterogeneity of $\{f_i^2\}_{i \in \mathcal{G}}$ is immediately obtained because $f_i^2 = f^2$ for every $i \in \mathcal{G} = \{1, \ldots, (1-\delta)n\}$, i.e., $|\nabla f_i^1(x) - \nabla f^2(x)| = 0 \leq \zeta_{\max}$.

Hence, we have shown that $\{f_i^1\}_{i \in \mathcal{G}}, \{f_i^2\}_{i \in \mathcal{G}} \in \mathcal{C}_{\mathrm{UH}}(\zeta_{\max})$.

Now, we show that

$$\frac{1}{2} \sum_{k=1}^2 \mathbb{E}_\pi |\nabla f^k(\mathcal{A}(\{f_{\pi(i)}^k\}_{i=1}^n))|^2 \geq \frac{\delta^2\zeta_{\max}^2}{4(1-\delta)} \tag{3}$$

If (3) is shown, we immediately have

$$\max_{k \in \{1,2\}} \mathbb{E}_\pi |\nabla f^k(\mathcal{A}(\{f_{\pi(i)}^k\}_{i=1}^n))|^2 \geq \frac{1}{2} \sum_{k=1}^2 \mathbb{E}_\pi |\nabla f^k(\mathcal{A}(\{f_{\pi(i)}^k\}_{i=1}^n))|^2$$

$$\geq \frac{\delta^2\zeta_{\max}^2}{4(1-\delta)} = \Omega(\delta^2\zeta_{\max}^2).$$

**Proof of (3)**  Let $\hat{x}_*^k$ be $\mathcal{A}(\{f_{\pi(i)}^k\}_{i=1}^n)$ for each $k \in \{1, 2\}$. Then, observe that

$$
\begin{aligned}
\mathbb{E}_\pi|\nabla f^1(\hat{x}_*^1)|^2 &= \mathbb{E}_\pi[|L\hat{x}_*^1 - \delta\zeta_{\max}/(1-\delta)|^2||\hat{x}_*^1| \le x_0]P(|\hat{x}_*^1| \le x_0) \\
&\quad + \mathbb{E}_\pi[|L(1_{\hat{x}_*^1 > x_0}x_0 + 1_{\hat{x}_*^1 < -x_0}(-x_0)) - \delta\zeta_{\max}/(1-\delta)|^2||\hat{x}_*^1| > x_0]P(|\hat{x}_*^1| > x_0) \\
&\ge \left( L^2\mathbb{E}_\pi[|\hat{x}_*^1|^2||\hat{x}_*^1| \le x_0] - \frac{2L\delta}{1-\delta}\zeta_{\max}\mathbb{E}_\pi[\hat{x}_*^1||\hat{x}_*^1| \le x_0] \right) P(|\hat{x}_*^1| \le x_0) \\
&\quad + \left( L^2|x_0|^2 - \frac{2L\delta}{1-\delta}\zeta_{\max}x_0 \right) P(|\hat{x}_*^1| > x_0) + \frac{\delta^2}{(1-\delta)^2}\zeta_{\max}^2 \\
&\ge \left( L^2|\mathbb{E}_\pi[\hat{x}_*^1||\hat{x}_*^1| \le x_0]|^2 - \frac{2L\delta}{1-\delta}\zeta_{\max}\mathbb{E}_\pi[\hat{x}_*^1||\hat{x}_*^1| \le x_0] \right) P(|\hat{x}_*^1| \le x_0) \\
&\quad + \left( L^2|x_0|^2 - \frac{2L\delta}{1-\delta}\zeta_{\max}x_0 \right) P(|\hat{x}_*^1| > x_0) + \frac{\delta^2}{(1-\delta)^2}\zeta_{\max}^2.
\end{aligned}
$$

For the last inequality, we used Jensen's inequality.

Also, we observe that

$$
\begin{aligned}
\mathbb{E}_\pi|\nabla f^2(\hat{x}_*^2)|^2 &= L^2\mathbb{E}_\pi[|\hat{x}_*^2|^2||\hat{x}_*^2| \le x_0]P(|\hat{x}_*^2| \le x_0) + L^2|x_0|^2P(|\hat{x}_*^2| > x_0) \\
&\ge L^2|\mathbb{E}_\pi[\hat{x}_*^2||\hat{x}_*^2| \le x_0]|^2P(|\hat{x}_*^2| \le x_0) + L^2|x_0|^2P(|\hat{x}_*^2| > x_0).
\end{aligned}
$$

$\mathcal{S}_n$ denotes the set of all the permutation on $[n]$. Now, let $\pi_{1\to2} \in \mathcal{S}_n$ be the permutation

$$
\begin{pmatrix} 1 & \dots & \delta n & \delta n + 1 & \dots & n \\ (1-\delta)n + 1 & \dots & n & 1 & \dots & (1-\delta)n \end{pmatrix}.
$$

Then, we can see that $\{f_{\pi(\pi_{1\to2}(i))}^1\}_{i=1}^n = \{f_{\pi(i)}^2\}_{i=1}^2$ for any random permutation $\pi \in \mathcal{S}_n$. Since $\pi \circ \pi_{1\to2}$ and $\pi$ follow the same distribution, we have $\mathbb{E}_\pi[q(\hat{x}_*^1)] = \mathbb{E}_\pi[q(\hat{x}_*^2)]$ for any function $q$. Then, by setting $\tilde{x} := \mathbb{E}_\pi[\hat{x}_*^1||\hat{x}_*^1| \le x_0] = \mathbb{E}_\pi[\hat{x}_*^2||\hat{x}_*^2| \le x_0]$ and $p := P(|\hat{x}_*^1| \le x_0) = P(|\hat{x}_*^2| \le x_0)$, it holds that

$$
\mathbb{E}_\pi|\nabla f^1(\hat{x}_*^1)|^2 \ge \left( L^2\tilde{x}^2 - \frac{2L\delta}{1-\delta}\zeta_{\max}\tilde{x} \right) p + \left( L^2|x_0|^2 - \frac{2L\delta}{1-\delta}\zeta_{\max}x_0 \right)(1-p) + \frac{\delta^2}{(1-\delta)^2}\zeta_{\max}^2
$$

and

$$
\mathbb{E}_\pi|\nabla f^2(\hat{x}_*^2)|^2 \ge L^2\tilde{x}^2 p + L^2|x_0|^2(1-p).
$$

Combining these results, we obtain

$$
\begin{aligned}
\frac{1}{2}\sum_{k=1}^2 \mathbb{E}_\pi|\nabla f^k(\hat{x}_*^k)|^2 &\ge \left( L^2\tilde{x}^2 - \frac{L\delta}{1-\delta}\zeta_{\max}\tilde{x} \right) p + \left( L^2x_0^2 - \frac{L\delta}{1-\delta}\zeta_{\max}x_0 \right)(1-p) + \frac{\delta^2}{2(1-\delta)^2}\zeta_{\max}^2 \\
&\ge \min_{\tilde{x}\in\mathbb{R}} \left( L^2\tilde{x}^2 - \frac{L\delta}{1-\delta}\zeta_{\max}\tilde{x} \right) + \frac{\delta^2}{2(1-\delta)^2}\zeta_{\max}^2.
\end{aligned}
$$

Finally, since $L^2\tilde{x}^2 - \frac{L\delta}{1-\delta}\zeta_{\max}\tilde{x}$ attains the minimum $-\delta^2\zeta_{\max}/(4(1-\delta)^2)$ at $\tilde{x} = \delta\zeta_{\max}/(2L(1-\delta))$, we obtain (3). This finishes the proof of Theorem 2.

## C  EFFICIENT IMPLEMENTATION OF SCREEN

Here, we provide an efficient implementation of Screen 2 introduced in Section 3. The concrete procedures are given by Algorithm 3. Observe that the expected running time is $\widetilde{O}(nd)$ rather than $O(n^2 d)$ when $K = \widetilde{O}(1)$. Theoretically, $K = \widetilde{O}(1)$ is sufficient to guarantee the best optimization error of $O(\delta^2 \zeta_{\max}^2)$. In our experiments in Section D.5, we used $s = 1.5$ and $K = 10$.

---

**Algorithm 3:** Efficient Implementation of Screen($\{m_i\}_{i=1}^n, \tau, s, K$)

---

1: # Server conservatively estimates the set of non-Byzantine workers.
2: # The expected runtime is only $\widetilde{O}(nd)$ when $K$ is $\widetilde{O}(1)$.
3: Set $\mathcal{J} = \emptyset$.
4: **for** $k = 1$ to $K$ **do**
5:     $j_{\text{ref}}^k = $ Search Reference Worker($\{m_i\}_{i=1}^n, \tau$).
6:     Append $j_{\text{ref}}^k$ to $\mathcal{J}$.
7: **end for**
8: Set $m_{\text{ref}} = \frac{1}{K} \sum_{k=1}^K m_{j_{\text{ref}}^k}$.
9: $\hat{\mathcal{G}} = \{i \in [n] : \|m_i - m_{\text{ref}}\| \leq (1 + s)\,\tau\}$.
10: $m := \frac{1}{|\hat{\mathcal{G}}|} \sum_{i \in \hat{\mathcal{G}}} m_i$
11: **Return:** $m$.

---

---

**Algorithm 4:** Search Reference Worker($\{m_i\}_{i=1}^n, \tau$)

---

1: # Serch worker $\hat{j}$ such that $|\{i \in [n] : \|m_i - m_{\hat{j}}\| \leq \tau\}| > n/2$.
2: # The expected runtime is only $O(nd)$.
3: **repeat**
4:     Randomly pick $\hat{j}$ from $[n]$.
5: **until** $|\{i \in [n] : \|m_i - m_{\hat{j}}\| \leq \tau\}| \leq n/2$
6: **Return:** $\hat{j}$.

---

## D    SUPPLEMENTARY OF NUMERICAL EXPERIMENTS

### D.1    DETAILS OF THE MODEL ARCHITECTURES

The architectures of FC and CNN were as follows:

- FC: Linear (output size: 100) → Softplus → Linear (output size: 10)
- CNN: Conv2d (# filters: 32, kernel size: 3) → ReLU → Conv2d (# filters: 64, kernel size: 3) → MaxPool2d (kernel size: 2) → Linear (output size: 128) → ReLU → Linear (output size: 10)
- VGG11: We used the official pytorch implementation of VGG11.[10]

We used the default initialization implemented in pytorch.

### D.2    DETAILS OF THE IMPLEMENTED ROBUST ALGORITHMS

Here, we provide the details of the implemented robust aggregations and their hyperparameter settings.

- Avg: The aggregation rule is defined as $\mathrm{Avg}(\{m_i\}_{i=1}^n) := (1/n)\sum_{i=1}^n m_i$.
- $\mathrm{CM}(\{m_i\}_{i=1}^n) := \{\mathrm{Med}(\{m_i|_j\}_{i=1}^n)\}_{j=1}^d$, where $\mathrm{Med}$ returns the one dimensional median of the input.
- KRUM (Blanchard et al., 2017): The aggregation rule is defined as $\mathrm{KRUM}(\{m_i\}_{i=1}^n) := \mathrm{argmin}_{m \in \{m_i\}_{i=1}^n} \min_{S \subset [n], |S| = (1-\delta_{\max})n} \sum_{j \in S} \|m - m_j\|^2$, where $\delta_{\max}$, which is an upper bound of $\delta$, was set to $1/4$[11].
- RFA (Pillutla et al., 2022): The aggregation rule is defined as $\mathrm{RFA}(\{m_i\}_{i=1}^n) := \mathrm{SmoothedWeiszfeld}(\{m_i\}_{i=1}^n, T, \nu)$, where the Smoothed Weiszfeld algorithm returns an approximate solution of the geometric median $\mathrm{argmin}_{m \in \mathbb{R}^d} \sum_{i=1}^n \|m - m_i\|$. $T$ and $\nu$ are the hyperparameters of Smoothed Weiszfeld and were set to 8 and 0.1 respectively.
- CClip Karimireddy et al. (2022): The aggregation rule is defined as $\mathrm{CClip}(\{m_i\}_{i=1}^n) := v + (1/n)\sum_{i=1}^n \min\{1, \tau/\|m_i - v\|\}(m_i - v)$, where $v$ is initialized as the previous aggregation result and this process is recursively executed for 3 iterations, where $\tau$ was set to 10. These hyperparameters were taken from the public source code of the authors of Karimireddy et al. (2022).
- MS (proposed): The aggregation rule is provided in Algorithm 1. We adopted Algorithm 2 rather than Algorithm 3 for simple implementations. Screening radius $\tau_t$ was set to $(\tau_\infty + (1 - \alpha/4)^{t-1}100^2)^{0.5}$, where $\tau_\infty$ was set to 20 for FC and VGG11 on MNIST, 30 for CNN on MNIST, 75 for FC and CNN on CIFAR10, and 100 for VGG11 on CIFAR10. For each iteration, $\tau_t$ was increased by 1.5 times from the original $\tau_t$ as long as $|\hat{\mathcal{G}}| < n/2$ in order to stabilize the performance.

### D.3    DETAILS OF THE IMPLEMENTED ATTACK ALGORITHMS

Here, we give the details of the five implemented attacks. In the following, $g_i^t$ denotes a stochastic gradient of worker $i$ used at iteration $t$.

- Bit Flipping (BF) attack: Byzantine worker $i$ sends $-g_i^t$ instead of $g_i^t$.
- Label Flipping (LF) attack: Label $y$ of the local datasets of Byzantine workers is flipped by $9 - y$ for every $y \in \{0, \ldots, 9\}$.
- Mimic attack (Karimireddy et al., 2022): Each Byzantine worker picks an honest worker $i_* \in \mathcal{G}$ and copies its output $g_{i_*}^t$ as $g_i^t$.

---

[10]https://pytorch.org/vision/main/models/generated/torchvision.models.vgg11.html.

[11]Note that this conservative choice of $\delta_{\max}$ is justified when KRUM is combined with bucketing (see Section C.2 of (Karimireddy et al., 2022)).

| Model/Data | AGG | BF | LF | Mimic | IPM | ALIE | Worst |
|---|---|---|---|---|---|---|---|
| CNN/ MNIST | Avg | $98.6 \pm 0.2$ | $98.6 \pm 0.2$ | $98.7 \pm 0.1$ | $\mathbf{98.5 \pm 0.1}$ | $57.6 \pm 66.0$ | $57.6 \pm 66.0$ |
| | CM | $98.0 \pm 0.4$ | $97.8 \pm 0.3$ | $98.1 \pm 0.4$ | $97.6 \pm 0.5$ | $76.4 \pm 73.0$ | $76.4 \pm 73.0$ |
| | KRUM | $97.6 \pm 0.1$ | $98.4 \pm 0.1$ | $98.0 \pm 0.3$ | $97.4 \pm 0.3$ | $\mathbf{98.5 \pm 0.2}$ | $97.4 \pm 0.3$ |
| | RFA | $98.6 \pm 0.1$ | $98.6 \pm 0.1$ | $98.6 \pm 0.1$ | $98.3 \pm 0.2$ | $83.7 \pm 41.9$ | $83.7 \pm 41.9$ |
| | CClip | $98.6 \pm 0.3$ | $98.6 \pm 0.1$ | $\mathbf{98.8 \pm 0.2}$ | $98.3 \pm 0.3$ | $49.6 \pm 86.3$ | $49.6 \pm 86.3$ |
| | MS (ours) | $\mathbf{98.7 \pm 0.2}$ | $\mathbf{98.7 \pm 0.2}$ | $98.7 \pm 0.2$ | $98.4 \pm 0.1$ | $98.4 \pm 0.2$ | $\mathbf{98.4 \pm 0.1}$ |
| CNN/ CIFAR10 | Avg | $\mathbf{68.7 \pm 0.9}$ | $\mathbf{69.6 \pm 0.6}$ | $\mathbf{69.1 \pm 0.8}$ | $68.6 \pm 1.0$ | $16.0 \pm 12.4$ | $16.0 \pm 12.4$ |
| | CM | $51.6 \pm 1.0$ | $52.1 \pm 2.0$ | $55.0 \pm 1.7$ | $46.8 \pm 3.3$ | $19.9 \pm 11.3$ | $19.9 \pm 11.3$ |
| | KRUM | $49.0 \pm 0.9$ | $52.3 \pm 2.0$ | $52.7 \pm 4.4$ | $42.9 \pm 1.4$ | $53.4 \pm 2.0$ | $42.9 \pm 1.4$ |
| | RFA | $66.7 \pm 1.0$ | $67.8 \pm 0.5$ | $67.9 \pm 1.6$ | $65.3 \pm 0.8$ | $18.6 \pm 11.4$ | $18.6 \pm 11.4$ |
| | CClip | $62.4 \pm 2.3$ | $64.0 \pm 0.7$ | $65.5 \pm 1.6$ | $51.7 \pm 1.1$ | $18.0 \pm 14.6$ | $18.0 \pm 14.6$ |
| | MS (ours) | $67.7 \pm 1.1$ | $67.4 \pm 1.8$ | $67.6 \pm 1.6$ | $65.6 \pm 1.7$ | $\mathbf{67.6 \pm 1.1}$ | $\mathbf{65.6 \pm 1.7}$ |

Table 2: Comparison of $95\%$ confidence intervals of the best test accuracy (%) against five attacks ($\delta = 3/20$) for CNN on each dataset ("Worst" shows the worst test accuracy among five attacks).

- Inner Product Manipulation (IPM) attack (Xie et al., 2020): Byzantine workers send $-(\varepsilon/|\mathcal{G}|) \sum_{i \in \mathcal{G}} g_i^t$, where $\varepsilon$ was set to 1.0 in our experiment.
- A Little Is Enough (ALIE) attack (Baruch et al., 2019): Byzantine workers compute the mean $\mu_\mathcal{G}$ and the standard deviation $\sigma_\mathcal{G}$ of $\{g_i^t\}_{i \in \mathcal{G}}$ and send $\mu_\mathcal{G} - z\sigma_\mathcal{G}$, where $z$ was set to 2.0 in our experiment.

### D.4 ADDITIONAL NUMERICAL RESULTS

Here, we provide additional numerical results not shown in the main paper due to space limitations.

#### D.4.1 ADDITIONAL NUMERICAL RESULTS OF EXPERIMENT 1

First, we provide the best accuracy comparison results on CNN (Table 2). From these results, we confirm that MS still outperformed the other methods for CNN.

Next, we report the learning curve of four metrics; train loss (Figures 2 and 5), train accuracy (Figures 4 and 5), test loss (Figures 6 and 7), and test accuracy (Figures 8 and 9) to complement the results of Experiment 1 in the main paper. The experimental settings were the same as in the main paper except that we added CM, KRUM, and RFA without bucketing.

#### D.4.2 ADDITIONAL NUMERICAL RESULTS OF EXPERIMENT 2

Here, we show the results of four metrics under the same setting as in the main paper: train loss, train acc, test loss, and test acc (Figures 10 and 11 to complement the results of Experiment 2.

### D.5 COMPARISON OF ALGORITHM 2 AND ALGORITHM 3

Here, we compare the performances of Algorithm 1 with Algorithm 3 to that with Algorithm 2.

The general settings of MS were the same as described in Section D.2 except for the settings of $\tau_\infty$; we used 15 for FC, CNN and VGG11 on MNIST, 50 for FC and CNN on CIFAR10, and 70 for VGG11 on CIFAR10. In Algorithm 3, we set $s = 1.5$ and $K = 10$. In the until loop, $\tau_t$ was increased by 1.1 times from the original $\tau_t$ if the termination condition is not satisfied for five randomly picked $\hat{j}$. After computing $m_{\text{ref}}$ by applying Algorithm 3 for $K$ times, we reset $\tau_t$ to the original value. Then, $\tau_t$ was increased by 1.1 times from the original $\tau_t$ as long as $|\hat{\mathcal{G}}| < n/2$ in order to stabilize the performance.

We ran MS_EFF (Algorithm 1 with Algorithm 3) in the settings of Experiment 1. The performance comparison is shown in Figures 12 and 13. From these results, we can see that MS_EFF still achieved the similar test accuracy to MS.

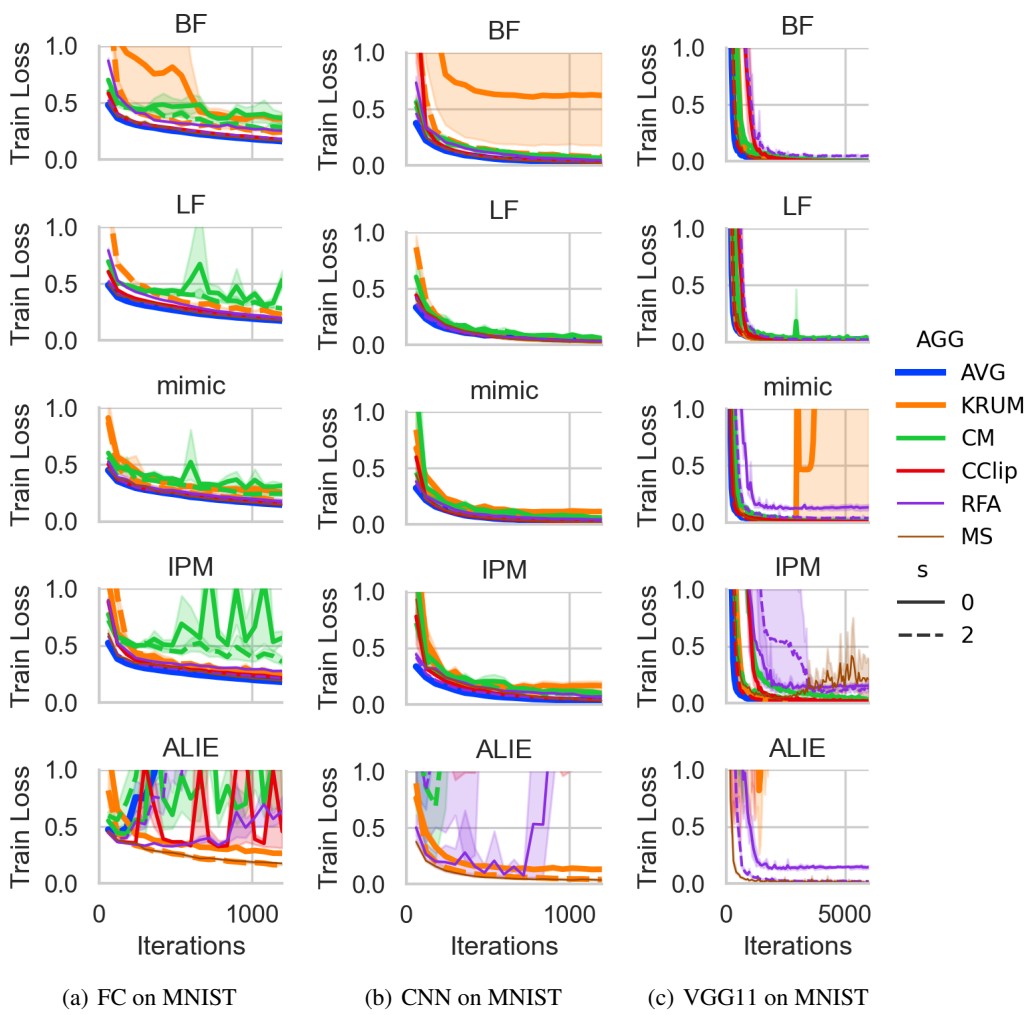

Figure 2: Comparison of the **train loss** of the implemented algorithms against the number of iterations for five attacks for three models on MNIST: BF, LF, mimic, IPM, and ALIE. $s$ denotes the bucketing size, and bucketing was not applied when $s = 0$.

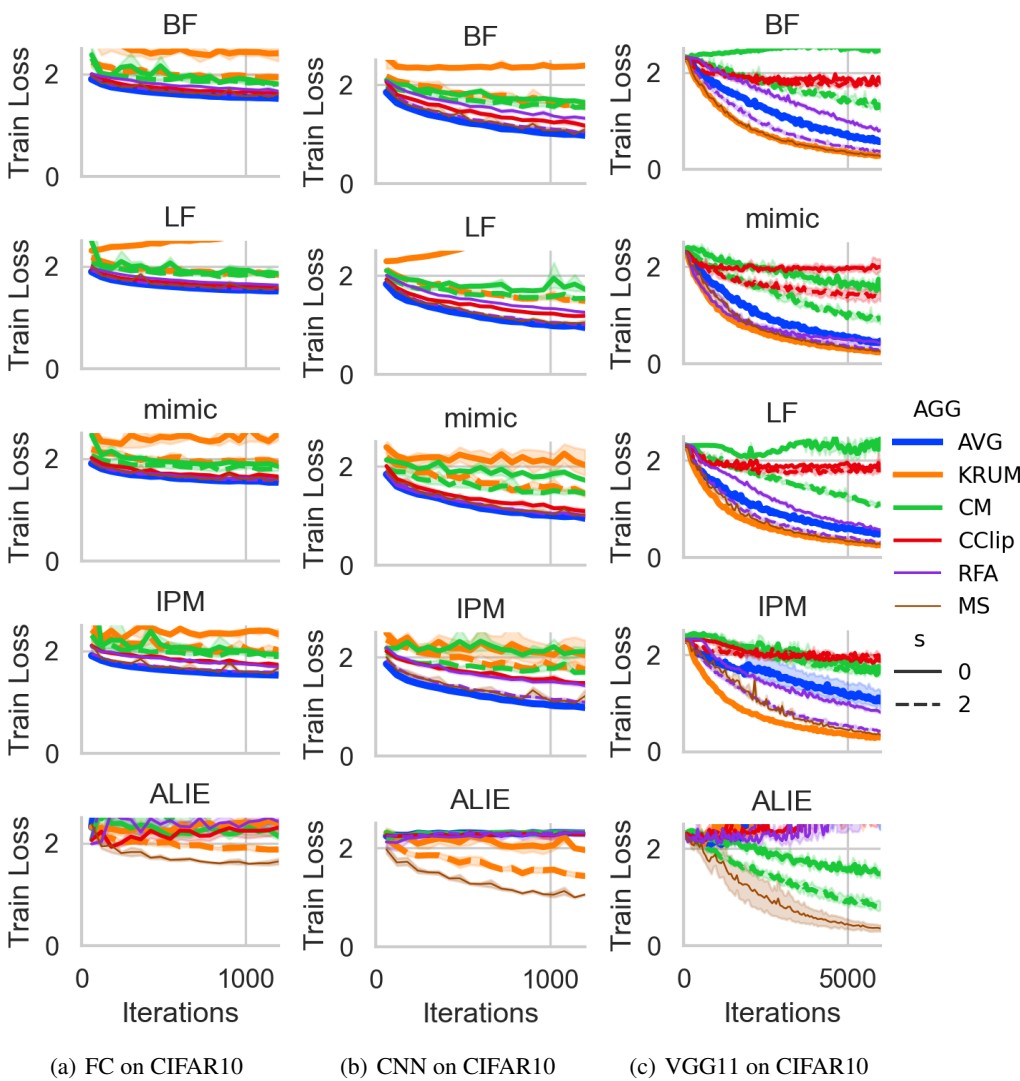

(a) FC on CIFAR10     (b) CNN on CIFAR10     (c) VGG11 on CIFAR10

Figure 3: Comparison of the **train loss** of the implemented algorithms against the number of iterations for five attacks for three models on CIFAR10: BF, LF, mimic, IPM, and ALIE. $s$ denotes the bucketing size, and bucketing was not applied when $s = 0$.

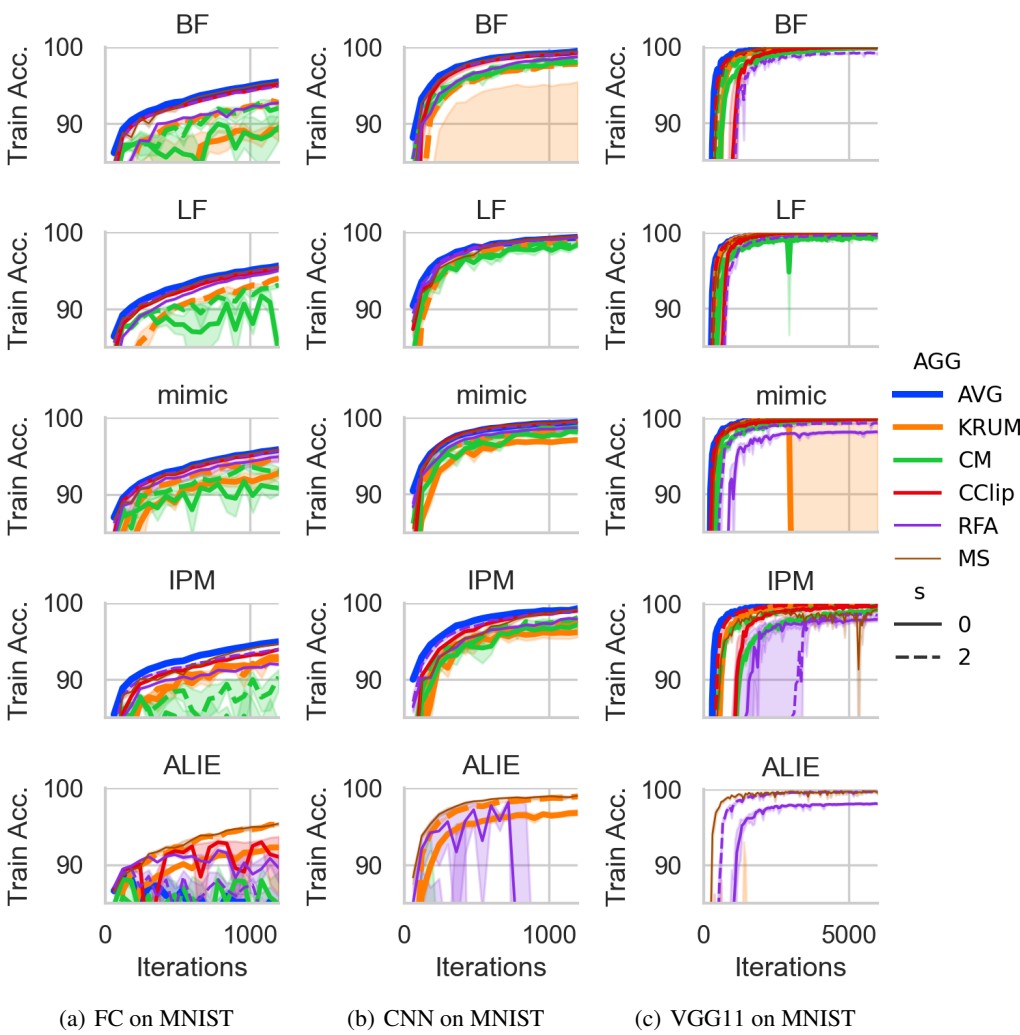

(a) FC on MNIST     (b) CNN on MNIST     (c) VGG11 on MNIST

Figure 4: Comparison of the **train accuracy** of the implemented algorithms against the number of iterations for five attacks for three models on MNIST: BF, LF, mimic, IPM, and ALIE. $s$ denotes the bucketing size, and bucketing was not applied when $s = 0$.

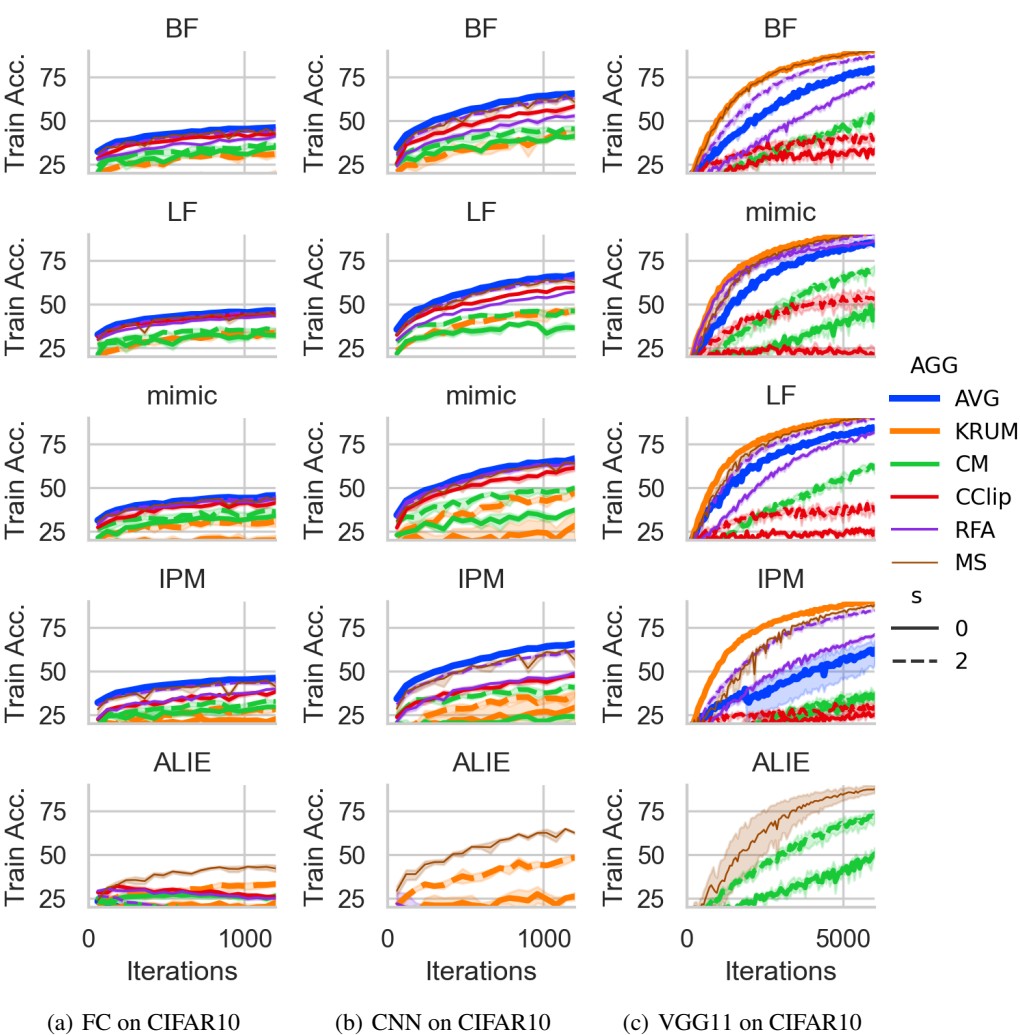

(a) FC on CIFAR10   (b) CNN on CIFAR10   (c) VGG11 on CIFAR10

Figure 5: Comparison of the **train accuracy** of the implemented algorithms against the number of iterations for five attacks for three models on CIFAR10: BF, LF, mimic, IPM, and ALIE. $s$ denotes the bucketing size, and bucketing was not applied when $s = 0$.

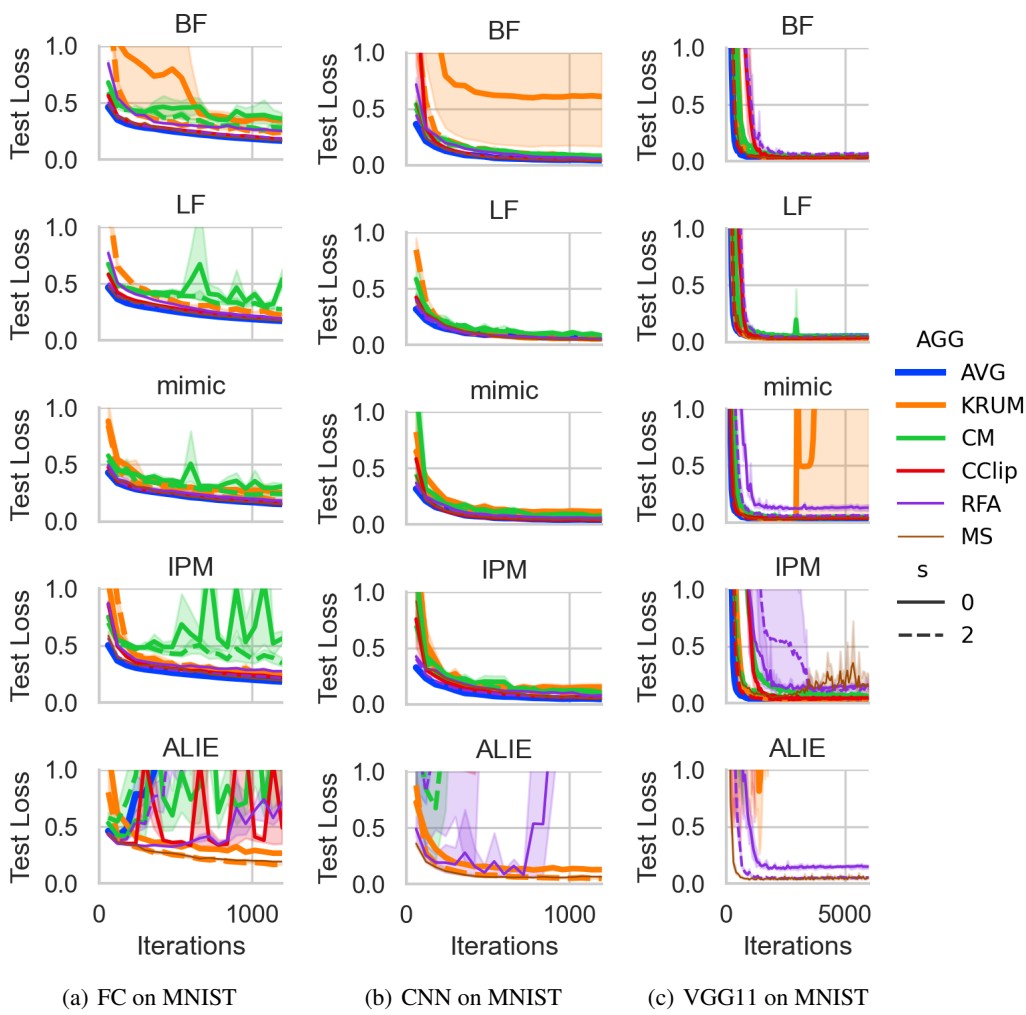

Figure 6: Comparison of the **test loss** of the implemented algorithms against the number of iterations for five attacks for three models on MNIST: BF, LF, mimic, IPM, and ALIE. $s$ denotes the bucketing size, and bucketing was not applied when $s = 0$.

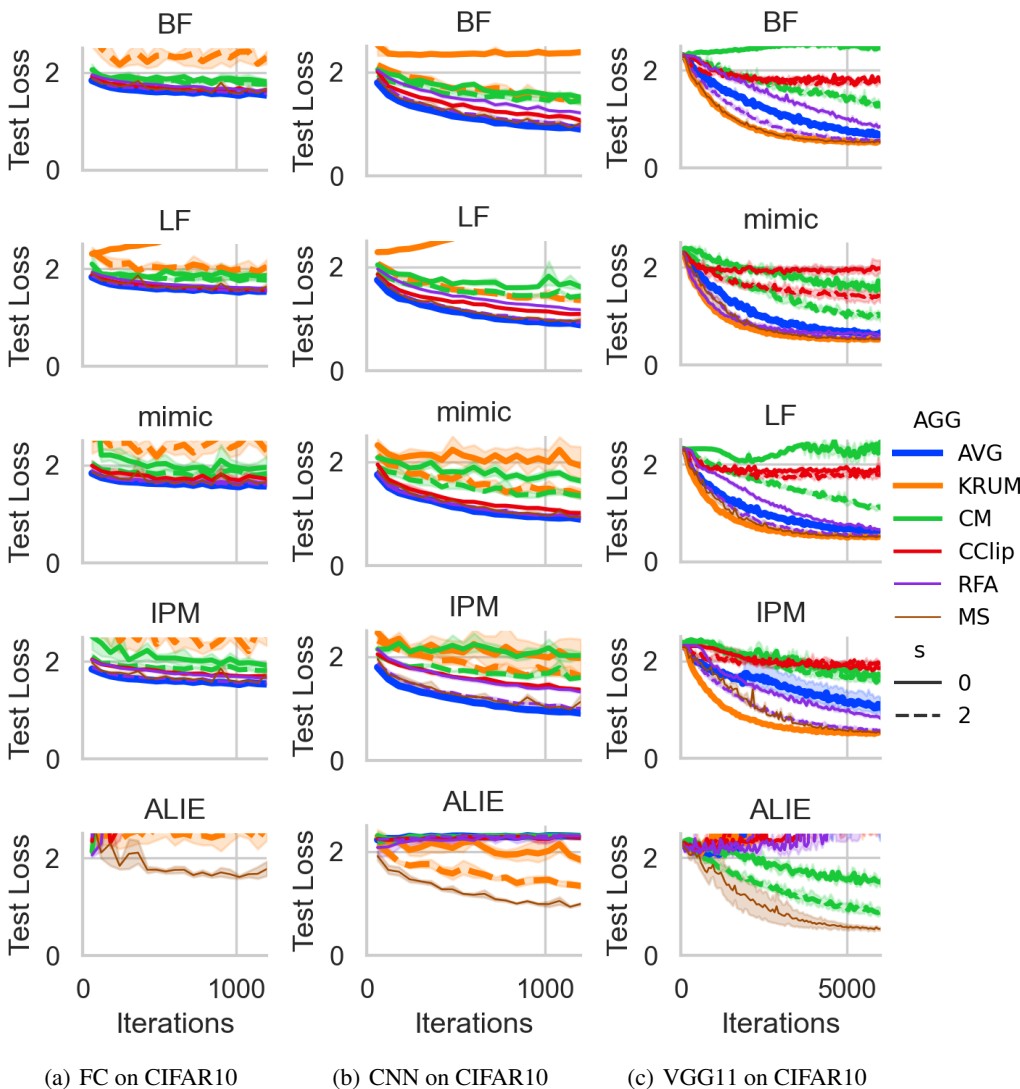

(a) FC on CIFAR10          (b) CNN on CIFAR10          (c) VGG11 on CIFAR10

Figure 7: Comparison of the **test loss** of the implemented algorithms against the number of iterations for five attacks for three models on CIFAR10: BF, LF, mimic, IPM, and ALIE. $s$ denotes the bucketing size, and bucketing was not applied when $s = 0$.

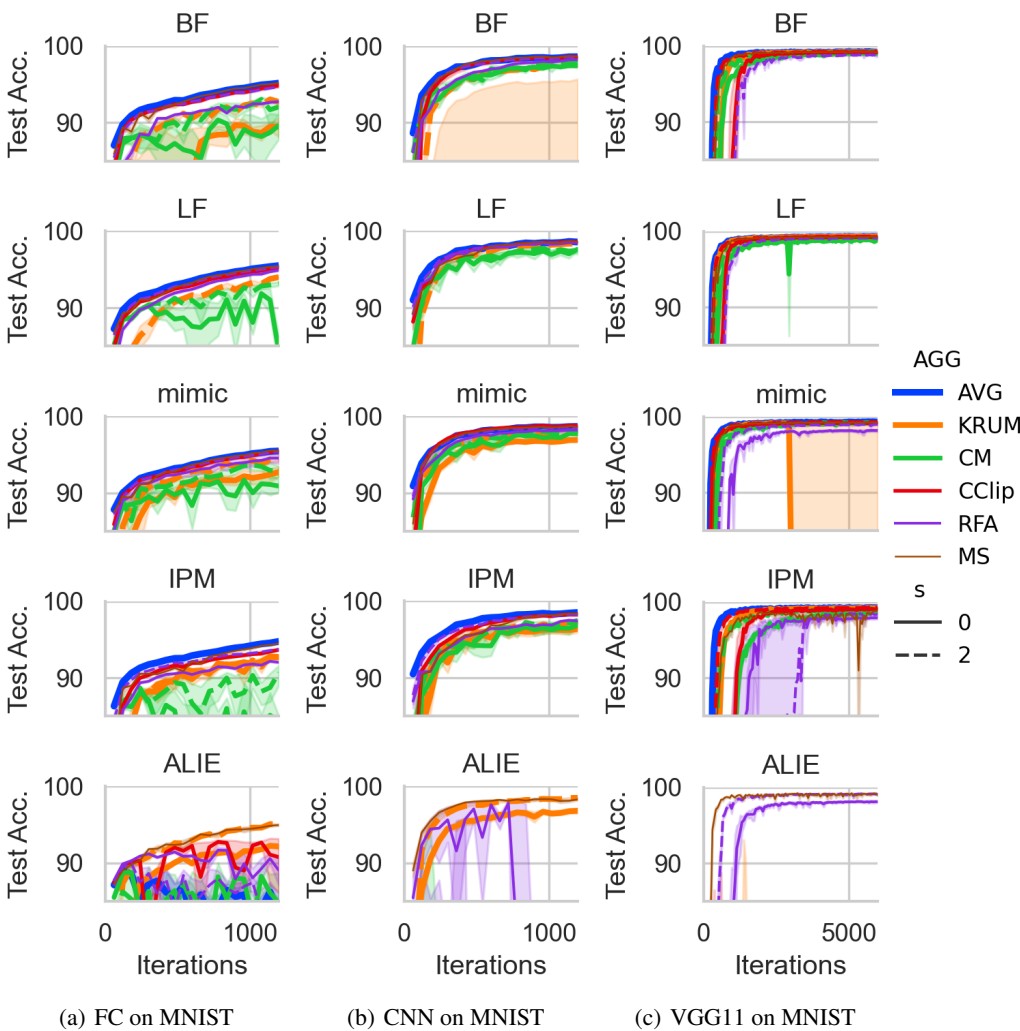

(a) FC on MNIST      (b) CNN on MNIST      (c) VGG11 on MNIST

Figure 8: Comparison of the **test accuracy** of the implemented algorithms against the number of iterations for five attacks for three models on MNIST: BF, LF, mimic, IPM, and ALIE. $s$ denotes the bucketing size, and bucketing was not applied when $s = 0$.

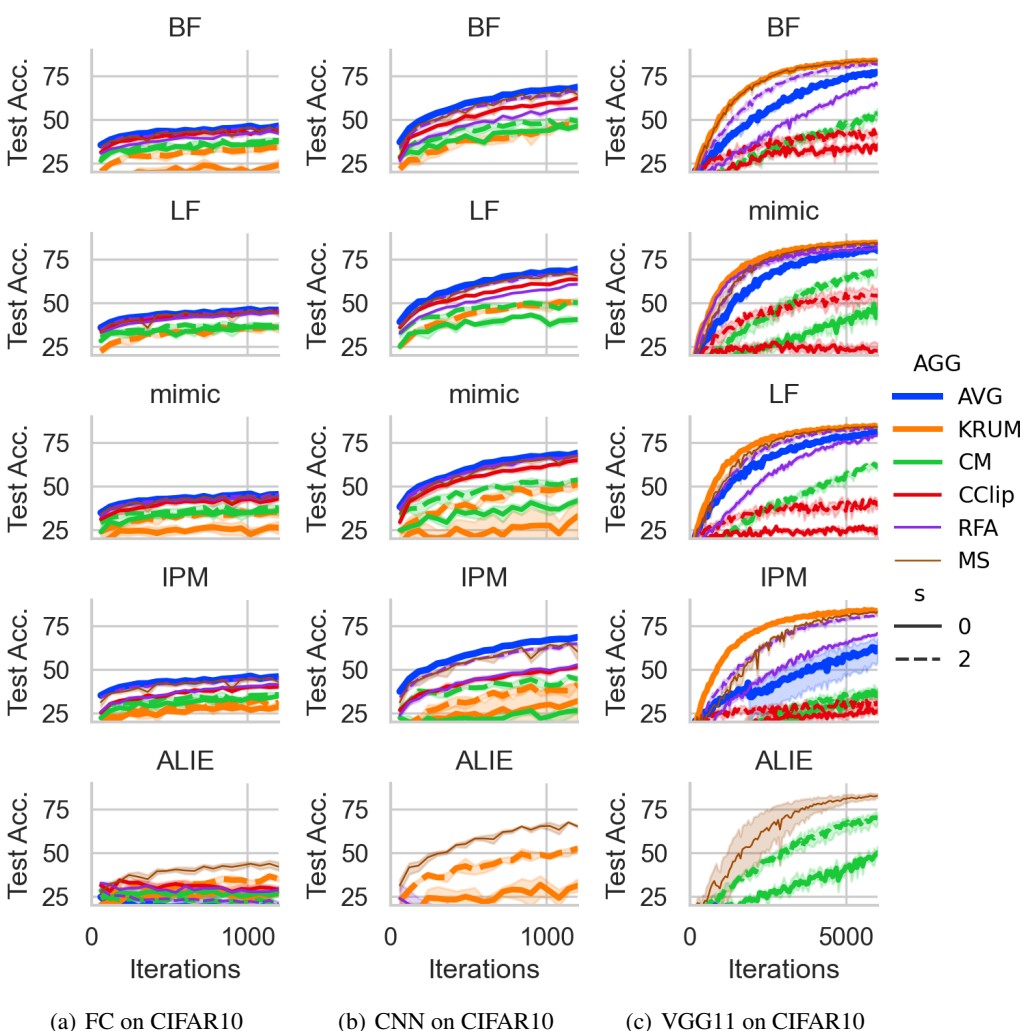

(a) FC on CIFAR10    (b) CNN on CIFAR10    (c) VGG11 on CIFAR10

Figure 9: Comparison of the **test accuracy** of the implemented algorithms against the number of iterations for five attacks for three models on CIFAR10: BF, LF, mimic, IPM, and ALIE. $s$ denotes the bucketing size, and bucketing was not applied when $s = 0$.

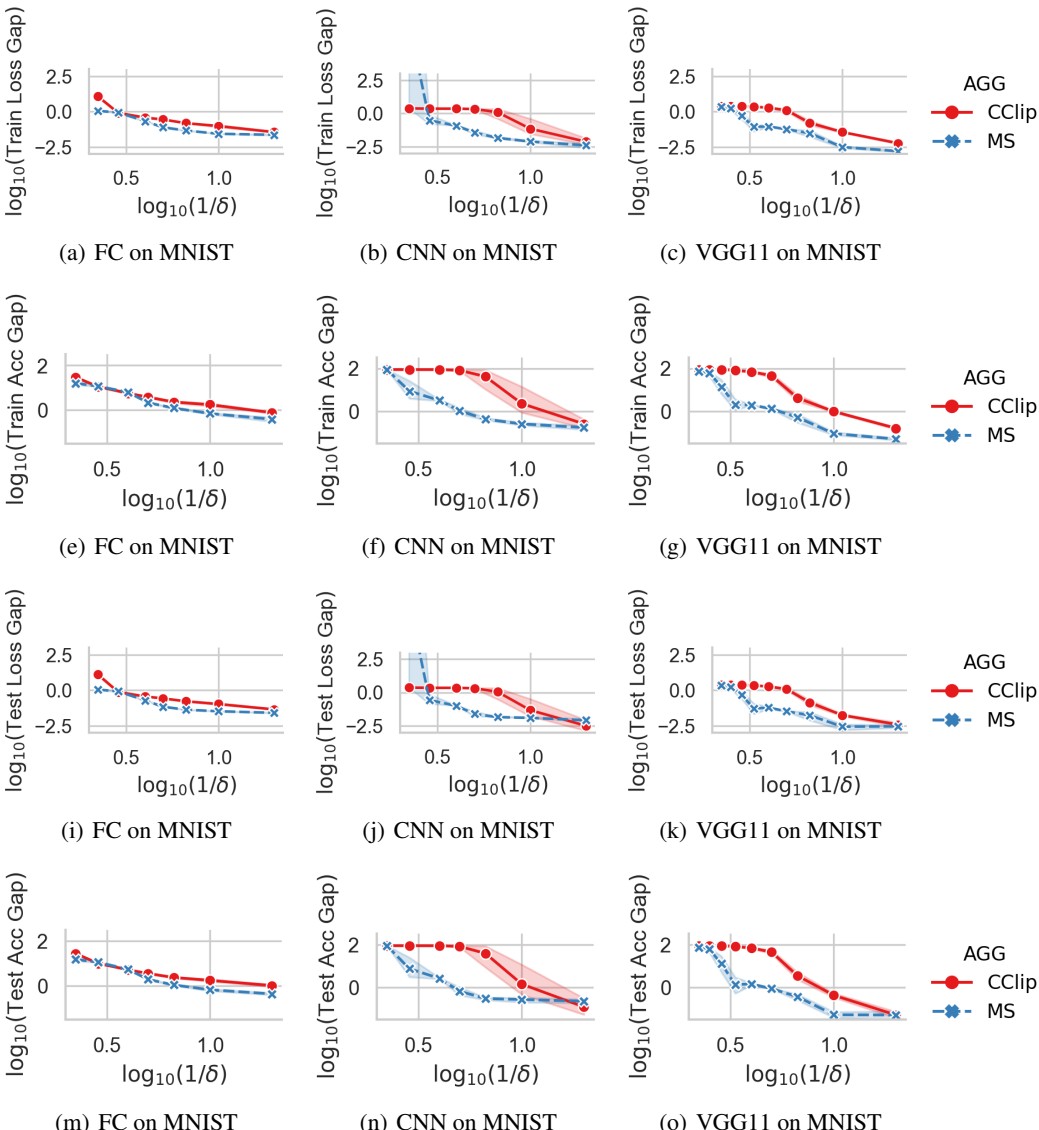

Figure 10: Comparison of the worst relative metrics to AVG without Byzantine workers (lower is better) of CClip and MS against the Byzantine fractions $\delta \in \{1/20, 2/20, 3/20, 4/20, 5/20, 7/20, 9/20\}$ for five attacks for three models on MNIST: BF, LF, mimic, IPM, and ALIE. (a)-(c) correspond to relative train loss, (e)-(g) to relative train accuracy, (i)-(k) to relative test loss, and (m)-(o) to relative test accuracy. The $x$-axis shows $1/\delta$ (i.e., the further to the right on the x-axis, the smaller $\delta$), and both axes are plotted on logarithmic scales.

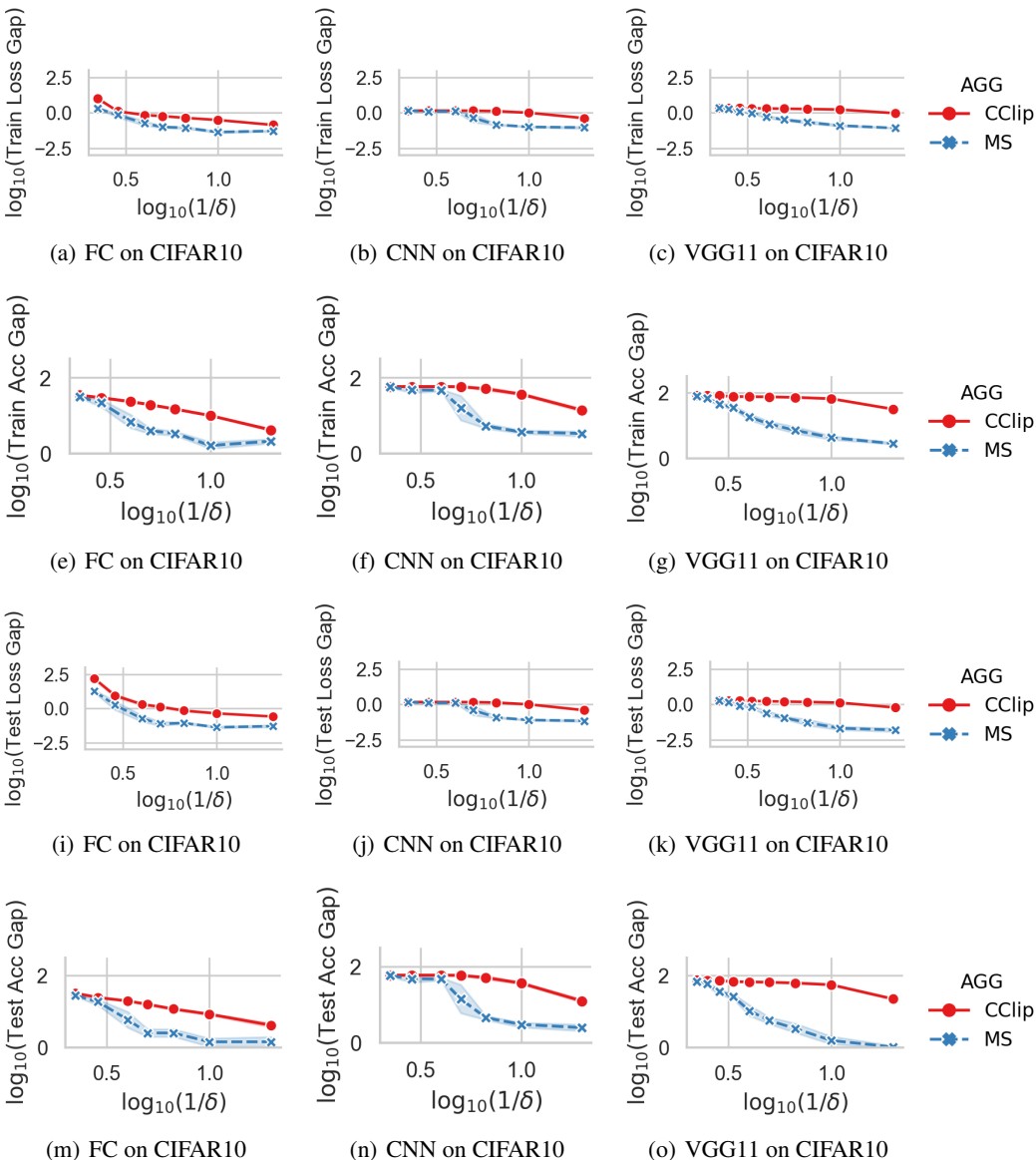

Figure 11: Comparison of the worst relative metrics to AVG without Byzantine workers (lower is better) of CClip and MS against the Byzantine fractions $\delta \in \{1/20, 2/20, 3/20, 4/20, 5/20, 7/20, 9/20\}$ for five attacks for three models on CIFAR10: BF, LF, mimic, IPM, and ALIE. (a)-(c) corresponds to relative train loss, (e)-(g) to relative train accuracy, (i)-(k) to relative test loss, and (m)-(o) to relative test accuracy. The $x$-axis shows $1/\delta$ (i.e., the further to the right on the x-axis, the smaller $\delta$), and both axes are plotted on logarithmic scales.

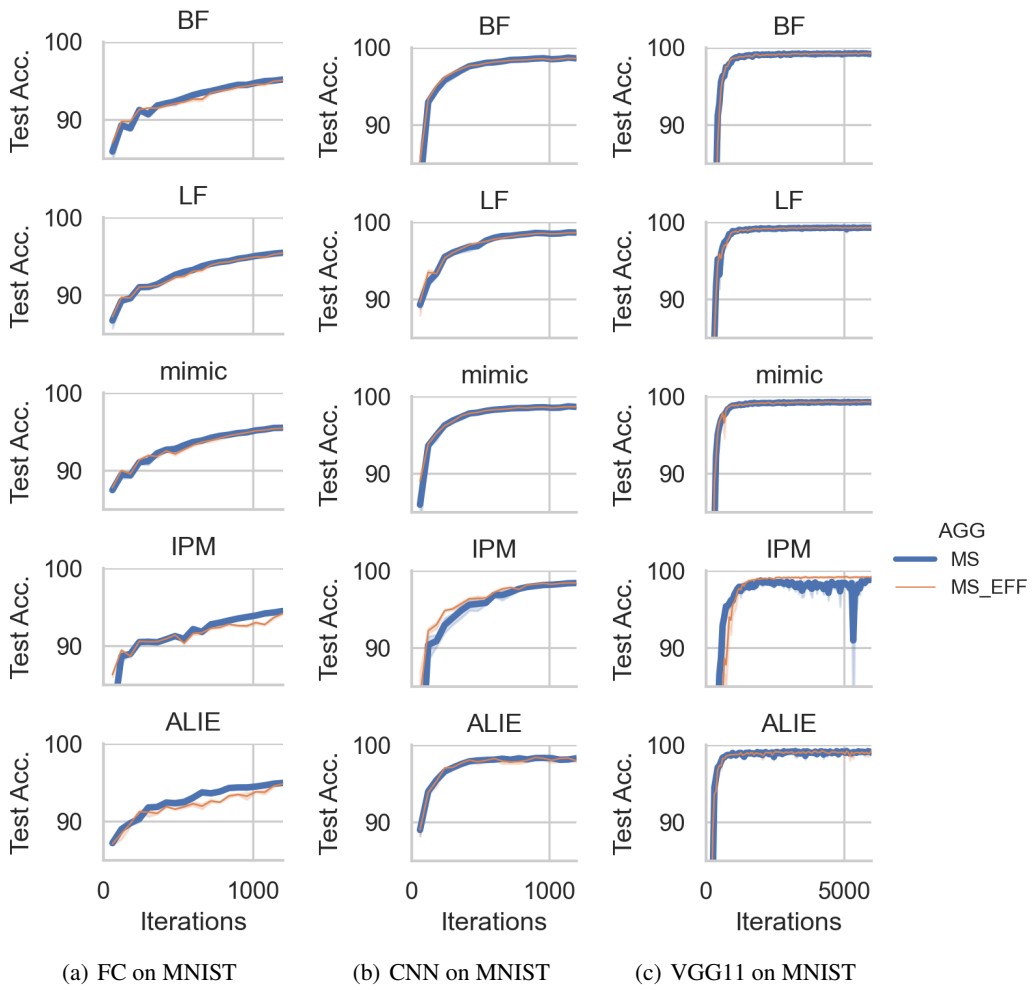

(a) FC on MNIST  (b) CNN on MNIST  (c) VGG11 on MNIST

Figure 12: Comparison of the **test accuracy** of MS_EFF (Algorithm 1 with Algorithm 3) to MS (Algorithm 1 with Algorithm 2) against the number of iterations for five attacks for three models on MNIST: BF, LF, mimic, IPM, and ALIE.

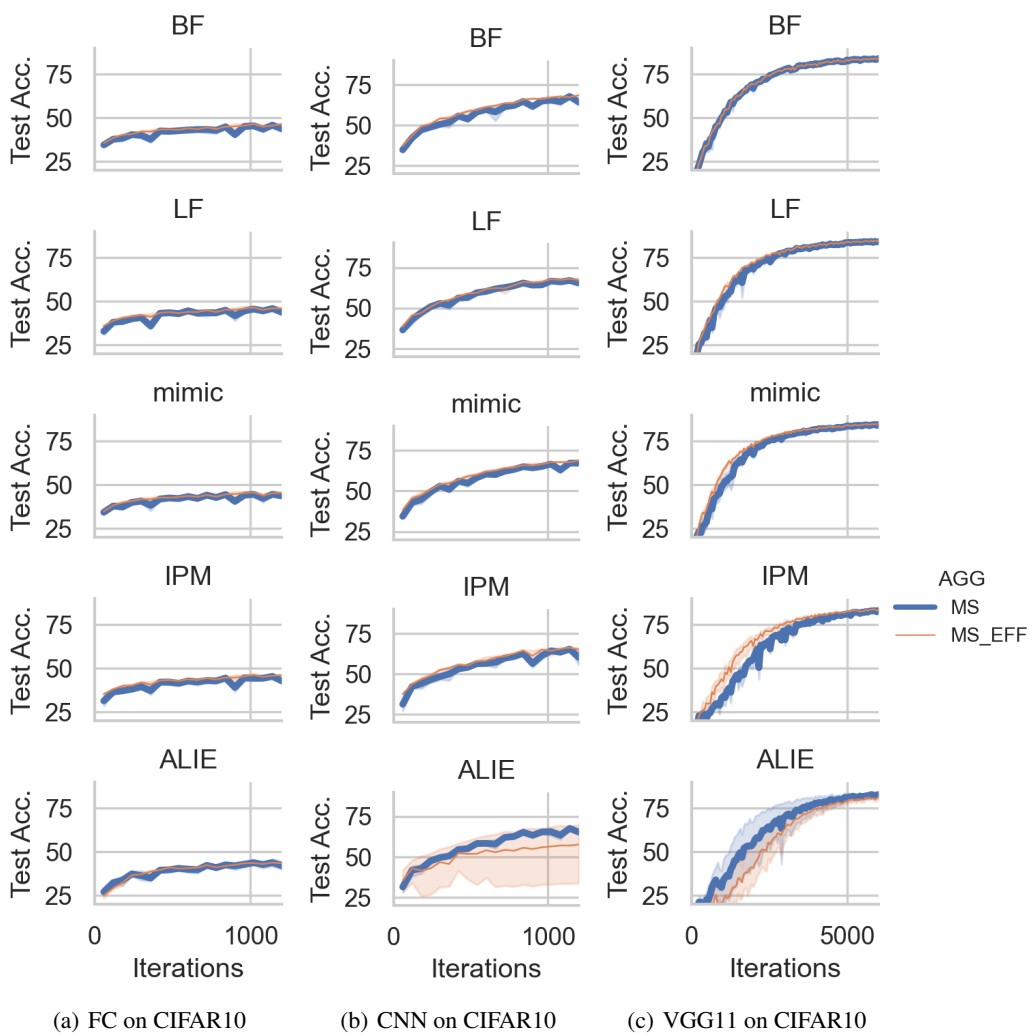

(a) FC on CIFAR10  (b) CNN on CIFAR10  (c) VGG11 on CIFAR10

Figure 13: Comparison of the **test accuracy** of MS_EFF (Algorithm 1 with Algorithm 3) to MS (Algorithm 1 with Algorithm 2) against the number of iterations for five attacks for three models on CIFAR10: BF, LF, mimic, IPM, and ALIE.

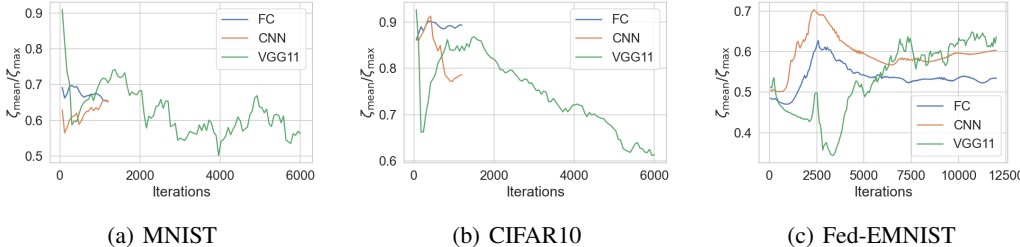

(a) MNIST  (b) CIFAR10  (c) Fed-EMNIST

Figure 14: Empirical values of $\zeta_{\mathrm{mean}}/\zeta_{\mathrm{max}}$ along the trajectories of momentum SGD without Byzantine workers for each model and dataset. On MNIST and CIFAR10, we ran 20 epochs $(1,200$ iterations) for FC and CNN, and 100 epochs $(6,000$ iterations) for VGG11. On Fed-EMNIST, we ran 200 epochs $(12,000$ iterations) for every model. Momentum parameter $\alpha = 0.1$, and learning rate $\eta = 0.01$ were commonly used. Minibatch size $b$ was fixed as 32 except for VGG11 on Fed-EMNIST. For VGG11 on Fed-EMNIST, we used a smaller batch size $b = 8$. [13]

# E  EMPIRICAL VALIDATIONS OF $\delta \leq (\zeta_{\mathrm{mean}}/\zeta_{\mathrm{max}})^2$

Here, we give empirical validations of the condition $\delta \leq (\zeta_{\mathrm{mean}}/\zeta_{\mathrm{max}})^2$, which is implicitly assumed in the main paper for the theoretical superiority of MS over the other existing methods, including CClip.

**Observations of $\zeta_{\mathrm{mean}}/\zeta_{\mathrm{max}}$ in the Scenarios of Section 6**  We report the observed values of $\zeta_{\mathrm{mean}}/\zeta_{\mathrm{max}}$ of local objectives $\{f_i\}_{i\in[P]}$ on MNIST and CIFAR10 in the settings of Section 6 using momentum SGD without Byzantine workers. We observed that $\zeta_{\mathrm{mean}}/\zeta_{\mathrm{max}}$ were not too small (about $0.50 \sim 0.93$) and $\delta \leq (\zeta_{\mathrm{mean}}/\zeta_{\mathrm{max}})^2$ was satisfied for $\delta \leq 1/4$. These results support the empirical superiority of our method over the other methods described in Section 6.

**Observations of $\zeta_{\mathrm{mean}}/\zeta_{\mathrm{max}}$ in a Real Federated Learning Dataset**  We further investigate the behavior of $\zeta_{\mathrm{mean}}/\zeta_{\mathrm{max}}$ of local objectives $\{f_i\}_{i\in[P]}$ on Fed-EMNIST [14], which is a *real federated learning dataset*. Due to memory constraints, we sampled 50 clients for training. We found that $\zeta_{\mathrm{mean}}/\zeta_{\mathrm{max}}$ was ranged in $0.34 \sim 0.70$ even with these settings, and these results suggest that *the condition $\delta \leq (\zeta_{\mathrm{mean}}/\zeta_{\mathrm{max}})^2$ would not be so restrictive even in practical federated learning.*

In summary, *the condition $\delta \leq (\zeta_{\mathrm{mean}}/\zeta_{\mathrm{max}})^2$ is not so restrictive from an empirical point of view.*

---

[11]https://github.com/TalwalkarLab/leaf

[13]When minibatch size $b = 32$ was used, we observed that SGD tended to approach a bad stationary point for VGG11 on Fed-EMNIST, which made the convergence quite slow. To avoid this phenomenon, we used a smaller batch size $b = 8$.

| Model/Data | AGG | BF | LF | Mimic | IPM | ALIE | Worst |
|---|---|---|---|---|---|---|---|
| FC/
Fed-MNIST | Avg | **66.5 ± 0.8** | **66.0 ± 0.8** | **66.0 ± 0.6** | **65.8 ± 0.7** | 38.7 ± 5.0 | 38.7 ± 5.0 |
| | CM | 41.8 ± 6.2 | 62.4 ± 1.4 | 60.4 ± 2.3 | 59.1 ± 0.9 | 38.4 ± 12.8 | 37.3 ± 10.8 |
| | KRUM | 16.2 ± 2.7 | 27.5 ± 5.0 | 19.4 ± 2.3 | 18.7 ± 5.4 | 17.2 ± 3.0 | 15.7 ± 2.4 |
| | RFA | 59.4 ± 4.0 | 65.7 ± 1.1 | 65.8 ± 1.7 | 64.6 ± 0.8 | 52.1 ± 7.2 | 52.1 ± 7.2 |
| | CClip | 22.4 ± 4.1 | 41.7 ± 6.9 | 27.4 ± 3.5 | 37.5 ± 3.2 | 21.8 ± 3.0 | 21.0 ± 3.1 |
| | MS (ours) | 65.4 ± 1.7 | 62.1 ± 1.8 | 64.5 ± 1.5 | 61.6 ± 1.5 | **65.0 ± 0.9** | **61.6 ± 1.5** |
| CNN/
Fed-EMNIST | Avg | 79.4 ± 0.3 | 79.0 ± 0.7 | **79.8 ± 1.4** | **79.7 ± 1.2** | 33.3 ± 29.8 | 33.3 ± 29.8 |
| | CM | 78.2 ± 0.9 | 77.7 ± 0.5 | 78.5 ± 1.3 | 77.5 ± 0.6 | 66.7 ± 9.7 | 66.7 ± 9.7 |
| | KRUM | 66.6 ± 2.3 | 67.6 ± 1.3 | 68.0 ± 0.7 | 69.2 ± 1.6 | 72.3 ± 2.3 | 66.4 ± 1.3 |
| | RFA | **79.5 ± 0.9** | 79.2 ± 1.1 | **79.8 ± 0.4** | 79.6 ± 1.4 | 65.5 ± 22.5 | 65.5 ± 22.5 |
| | CClip | 79.3 ± 1.2 | **79.6 ± 0.8** | 79.7 ± 1.0 | 71.3 ± 30.0 | **79.9 ± 1.3** | 71.3 ± 30.0 |
| | MS (ours) | 78.8 ± 0.6 | 76.8 ± 1.1 | 78.3 ± 2.0 | 74.5 ± 5.1 | 78.9 ± 1.7 | **74.3 ± 4.6** |
| VGG11 /
Fed-MNIST | Avg | 80.6 ± 1.8 | 81.0 ± 1.3 | 81.4 ± 1.3 | **82.0 ± 2.1** | 15.5 ± 9.5 | 15.5 ± 9.5 |
| | CM | 80.4 ± 1.2 | 80.6 ± 2.0 | 80.7 ± 1.3 | 81.0 ± 1.9 | 28.2 ± 9.1 | 28.2 ± 9.1 |
| | KRUM | 68.4 ± 2.4 | 71.5 ± 0.5 | 72.9 ± 1.2 | 71.2 ± 1.5 | 76.4 ± 1.1 | 68.4 ± 2.4 |
| | RFA | 80.7 ± 1.7 | 82.0 ± 1.9 | **81.8 ± 1.4** | 81.9 ± 0.8 | 42.2 ± 7.9 | 42.2 ± 7.9 |
| | CClip | **81.7 ± 1.1** | **82.5 ± 1.5** | 81.5 ± 1.0 | 80.6 ± 1.6 | 64.9 ± 3.5 | 64.9 ± 3.5 |
| | MS (ours) | 79.4 ± 2.9 | 81.7 ± 1.8 | 80.8 ± 2.2 | 78.3 ± 1.7 | **80.7 ± 1.7** | **78.2 ± 1.4** |

Table 3: Comparison of 95% confidence intervals of the best test accuracy (%) against five attacks ($\delta = 5/50$) for FC, CNN, and VGG11 on Fed-EMNIST ("Worst" shows the worst test accuracy among five attacks).

# F  ACCURACY COMPARISON ON FED-EMNIST

In this section, we provide empirical comparison of our method with the existing methods on Fed-EMNIST, which is a 62-class classification dataset for a realistic federated learning scenario.

## F.1  EXPERIMENTAL SETTINGS

Due to the memory constraints, we sampled 50 clients for training, and 10 clients for testing. We ran 200 epochs (12,000 iterations) for every model. Momentum parameter $\alpha = 0.1$, and learning rate $\eta = 0.01$ were commonly used. Minibatch size $b$ was fixed as 32 for FC and CNN, and was fixed as 8 for VGG11. For MS, we used $\tau_\infty = 75$ for FC and CNN, and $\tau_\infty = 50$ for VGG11. The number of Byzantine workers was set to 5, i.e., $\delta = 5/50 = 0.1$. The other experimental settings were the same as in Section 6.

## F.2  EXPERIMENTAL RESULTS

Table 3 shows the best accuracy comparison results on Fed-EMNIST. From Table 3, we can see that MS consistently outperformed the other methods in terms of the worst best test accurary against the five attacks.

