# OpenReview forum: "Simple Minimax Optimal Byzantine Robust Algorithm for Nonconvex Objectives with Uniform Gradient Heterogeneity"
_ICLR.cc/2024/Conference — ICLR 2024 poster_

### Official Review · Reviewer_uTUe · 2023-10-26

**Soundness:** 3 good
**Presentation:** 2 fair
**Contribution:** 3 good
**Rating:** 6
**Confidence:** 3

**Summary:**

This paper proposes a new Byzantine robust algorithm called Momentum Screening (MS) for nonconvex federated learning. MS uses a simple screening test to detect and remove potentially malicious gradients from Byzantine workers. The remaining gradients are aggregated using standard momentum SGD. The algorithm is adaptive to the Byzantine fraction $δ$.

This paper gives theoretical analysis on the proposed algorithm, showing that MS achieves an optimization error of O($δ^2 ζ^2_{max}$) for the class $C_{UH}(\zeta_{max})$. A matching minimax lower bound is also provided for $C_{UH}(\zeta_{max})$. This rate differs from the rate $O(\delta \zeta_{mean}^2)$ for the class $C_{MH}(\zeta_{mean})$.

Experiments on MNIST and CIFAR10 with neural nets demonstrate MS outperforms existing methods like Centered Clipping, especially for small $δ$. This aligns with the better dependence on $δ$.

**Strengths:**

1. The proposed algorithm (MS) is simple to implement yet performs well. It also does not need to know the Byzantine fraction $\delta$ in advance, which is practical.
1. The rate of convergence of MS is better than the previous best known rate of $O(δζ^2_{mean})$ under the (corrected) condition when $\delta \leq (\zeta_{mean}/\zeta_{max}/)^2$, so the method is preferred if Byzantine workers are very few.
1. The author also provide a computationally efficient algorithm for their proposed MS method, whose performance is only worse than the original one by a constant factor.
1. Some experiments on MNIST and CIFAR10 with neural nets shows that MS outperforms other method.

**Weaknesses:**

1. In the literature the rate $O(δζ^2_{mean})$ is derived from $C_{MH}(\zeta_{mean})$, while this paper gives the rate O($δ^2 ζ^2_{max}$) for $C_{UH}(\zeta_{max})$. Now, to give a better rate, one needs
$$δ^2 ζ^2_{max} \leq δζ^2_{mean}  \Leftrightarrow \delta \leq (ζ_{mean} / ζ_{max})^2, $$
where the RHS is $\leq 1$ since $ ζ_{max} \geq ζ_{mean}$.
Therefore, **the requirement of $\delta$ is wrong throughout the paper** (the authors give $\delta \leq ( ζ_{max}/ ζ_{mean})^2$).
The authors even did not notice this mistake when they write $\delta = \Omega(1)$ (in Section 7) but in fact Byzantine fraction  $\delta < 0.5$.
Such mistake makes me doubt the correctness of the proof in this paper, but I do not have enough time to check the whole proof.

1. As argued in this paper, $ ζ_{max} \gg ζ_{mean} $, meaning that the method is only favourable when $\delta$ is very small, which seems to be not practical in the Byzantine workers setting. Moreover, since $C_{UH}(\zeta_{max})$ and $C_{MH}(\zeta_{mean})$ are different hypothesis classes, directly comparing rates seems to be improper. An analysis of MS in $C_{MH}(\zeta_{mean})$ is also needed.

1. Although the hyperparameter $\tau_t$ is adaptive to the Byzantine fraction $\delta$, it has to be be chosen according to $\zeta_{max}$, which is unknown in priori, so an inproper choice of $\tau$ could harm the performance of the algorithm.
It would be favourable to provide an empirical way to choose $\tau_t$.

1. For the presentation of the paper, it would be clearer if the author provides a sketch of the proof rather than presenting directly some propositions.

**Questions:**

1. Could the authors comment more on the relation between $\zeta_{max}$ and $\zeta_{mean}$, particularly with some real datasets?

---

> ### Author Response · Authors · 2023-11-15
> **Reply to Reviewer uTUe**
>
> Thank you for your important comments.
>
> **About Weakness 1**.
>
> As you pointed out, the expression $\delta \leq (\zeta_\mathrm{max}/\zeta_\mathrm{mean})^2$ should be fixed as $\delta \leq (\zeta_\mathrm{mean}/\zeta_\mathrm{max})^2$ (please check the revision paper). We thank you for your pointing out these typos and apologize for them, but, they are simple systematic typos and do not affect the proofs given in Sections A and B.
>
> **About Weakness 2 and Question 1**
>
> We believe that $\delta \leq (\zeta_\mathrm{mean}/\zeta_\mathrm{max})^2$ is reasonable because ***$\zeta_\mathrm{mean}/\zeta_\mathrm{max}$ is not too small in practice***. In fact, after reading your review, we conducted additional experiments to address the concern that $\zeta_\mathrm{mean}/\zeta_\mathrm{max}$ is very small in practice. Specifically, we examined the empirical values of $\zeta_\mathrm{mean}/\zeta_\mathrm{max}$ on the settings of Section 6 and additionally on Fed-EMNIST, which is a real FL dataset. The experimental results can be found in Section E of the revised paper (p.36). From Figure 14 in Section E, we can see that ***$\zeta_\mathrm{mean}/\zeta_\mathrm{max}$ was in $0.5 \sim 0.93$ on MNIST and CIFAR10, and was in $0.34 \sim 0.70$ on Fed-EMNIST***. From these observations, we conclude that the condition ***$\delta \leq (\zeta_\mathrm{mean}/\zeta_\mathrm{max})^2$ is practical enough and the benefits of MS are not so limited***.
>
> > Moreover, since $\mathcal C_\mathrm{UH}(\zeta_\mathrm{max})$
>  and $\mathcal C_\mathrm{MH}(\zeta_\mathrm{mean})$
>  are different hypothesis classes, directly comparing rates seems to be improper. An analysis of MS in $\mathcal C_\mathrm{MH}(\zeta_\mathrm{mean})$
>  is also needed.
>
> We disagree with the first point. In general, it is very common in the machine learning and optimization literature to show that an algorithm achieves a better rate (generalization error, optimization error, etc.) for a smaller hypothesis class than the rate of another algorithm targeting a larger hypothesis class. For example, it is well known that a typical theory of LASSO shows that LASSO achieves a better generalization error than OLS when the true parameter is sparse, where LASSO assumes a smaller hypothesis class than OLS does.
>
> Regarding the second point, we agree that an analysis of MS in $\mathcal C_\mathrm{MH}(\zeta_\mathrm{mean})$ may be necessary to judge which algorithm is better in $\mathcal C_\mathrm{MH}(\zeta_\mathrm{mean})$. However, our main focus is to construct a minimax optimal algorithm for the hypothesis class $\mathcal C_\mathrm{UH}(\zeta_\mathrm{max})$, and as shown in Figure 14 in Section E of the revised paper, it is empirically justified to assume that the local objectives are in $\mathcal C_\mathrm{UH}(\zeta_\mathrm{max})$ for a reasonable $\zeta_\mathrm{max}$ because $\zeta_\mathrm{max} \not \gg \zeta_\mathrm{mean}$ empirically. Thus, this point does not detract from the importance of this study.
>
> **About Weakness 3**.
>
> In our experiments, as described in Section D.2, we used a heuristic strategy to stabilize the performance that $\tau_t$ was increased by $1.5$ times from the original $\tau_t$ as long as $|\hat {\mathcal G}| < n/2$, although  $\tau_\infty$, which is the limit of $\tau_t$ ($t \to \infty$) , had to be roughly tuned. As you said, it is practically desirable that the algorithm be adaptive not only to Byzantine fraction $\delta$ but also to heterogeneity $\zeta_\mathrm{max}$ (or $\zeta_\mathrm{mean}$). This direction is definitely an important future work.
>
> **About Weakness 4**.
>
> Thanks for your suggestion. Actually, we give a brief overview of the analysis in the first part of Section 4 to clarify our proof strategy. If that is not enough, we would like to add a more detailed sketch of the proof.
>
>
> We would be very happy if your concerns were addressed and the score would be raised.

---

> > ### Comment · Reviewer_uTUe · 2023-11-22
> >
> > Thank you for your response.
> > It clarifies my particular concern on the quantity $\zeta_{mean} / \zeta_{max}$.
> > I would like to raise my score.

---

### Official Review · Reviewer_BeSt · 2023-11-01

**Soundness:** 3 good
**Presentation:** 3 good
**Contribution:** 3 good
**Rating:** 6
**Confidence:** 3

**Summary:**

This paper considers the problem of federated learning with Byzantine workers who can send arbitrary responses to the central server. In the non-IID case where the local distributions of non-Byzantine workers are heterogeneous, the standard aggregations will fail empirically, as shown in previous works. In this paper, the authors developed a new, simple byzantine robust algorithm that have better minimax optimal optimization error compared to the best previous algorithm when the maximum gradient heterogeneity is not much larger than the average gradient heterogeneity, whose optimality in this parameter regime is demonstrated by establishing a lower bound result. Moreover, the authors conducted numerical experiments to support their theoretical analysis.

**Strengths:**

The algorithm is novel and simple, makes it relatively easy to be implemented in practice. Moreover, the improvement in the minimax optimal optimization error is significant in the parameter regime where the maximum gradient heterogeneity is around the same order as the average gradient heterogeneity, which seems like a common assumption in various practical situations. The performance of the algorithm is also well demonstrated in the various numerical experiments.

**Weaknesses:**

The convergence rate in terms of the number of steps $T$ might not be optimal. In particular, the algorithm is a momentum-based method, however, the convergence rate exhibits the form of a non-momentum based method, and it is unclear to me why the momentum is needed here.

**Questions:**

Will the convergence rate of the algorithm remain unchanged if the momentum is removed? Or, is there a better momentum-based algorithm that has better convergence rate?

---

> ### Author Response · Authors · 2023-11-15
> **Reply to Reviewer BeSt**
>
> Thank you for your insightful comments and questions.
>
> > The convergence rate in terms of the number of steps
>  might not be optimal. In particular, the algorithm is a momentum-based method, however, the convergence rate exhibits the form of a non-momentum based method, and it is unclear to me why the momentum is needed here.
>
> > Will the convergence rate of the algorithm remain unchanged if the momentum is removed? Or, is there a better momentum-based algorithm that has better convergence rate?
>
> Our algorithm relies on a heavy-ball method rather than the famous Nesterov's acceleration method. Thus, it is natural that the convergence rate matches that of non-momentum based methods when $\delta \to 0$, since a heavy-ball method does not improve the convergence rate of non-momentum methods at least in the standard nonconvex optimization theory (see, for example,  Mai and Johansson, 2020 [1]). Although the main focus of this paper is on the asymptotic optimization error, obtaining the optimal convergence rate in terms of the number of steps $T$ based on Nesterov's acceleration is an interesting future direction.
>
> The reason why we introduce the momentum in our algorithm is that ***the momentum mitigates the effect of the stochastic noise*** by canceling out the noise thanks to the accumulation of the previous stochastic gradients. This is very important when $\delta > 0$ because the screening algorithm judges the workers to be non-Byzantine or Byzantine based on comparing the distance between the workers' outputs. If the momentum is removed, the screening algorithm must aggregate the stochastic gradients only at the current iteration, which will degrade the performance of the distance-based detection of the Byzantine workers due to the large  stochastic noise. As a result, ***the convergence rate will be degraded when the momentum is removed***.
>
> [1] Mai and Johansson, 2020:  Convergence of a Stochastic Gradient Method with Momentum
> for Non-Smooth Non-Convex Optimization.

---

> > ### Comment · Reviewer_BeSt · 2023-11-22
> >
> > I would like to thank the authors for answering my questions, and I remain my rating.

---

### Official Review · Reviewer_p51w · 2023-11-06

**Soundness:** 3 good
**Presentation:** 2 fair
**Contribution:** 2 fair
**Rating:** 3
**Confidence:** 2

**Summary:**

The paper studies nonconvex federated learning (FL) in the presence of byzantine workers with a fraction of  $\delta$ out of the workers. Then the authors proposed the Momentum Screening (MS) algorithm for such setting, achieving $O(\delta^2 \zeta^2_{max})$ error rate for $\zeta_{max}$-uniform gradient heterogeneity, and showed the minimax optimality of the proposed method in such setting. Experimental results are then given to validate the MS algorithm.

**Strengths:**

The algorithmic structure of the MS algorithm is simple and can adapt to the Byzantine fractions $\delta$, all of which can be practically attractive. Furthermore, the minimax optimality results seem like the first of its kind for such setting of $\zeta_{max}$-uniform gradient heterogeneity.

**Weaknesses:**

1. The consideration of algorithmic design for uniform gradient heterogeneity as in this paper has been done in the literature. In fact, the rate achieved here seems to be the same as the CCLIP method (Karimireddy et al. (2022)) (ref [1] as below for convenience). Yet, such literature was not well discussed enough in the paper.
2. Following up the above point, many results in the paper are quite the same as those in CCLIP without improvement, and the analysis is quite natural and motivated from previous work. The true technical novelty of the paper, besides the MS method with simplicity, is perhaps the fact that they proved lower bound in the minimax sense for uniform gradient heterogeneity. However, this is quite a natural extension from the first work that proved such results for the case of mean gradient heterogeneity.
3. Systematic typo throughout the paper: note that yours is better than CCLIP when $\delta \leq ( \zeta_{mean}/ \zeta_{max})^2$. Can you give a sense of what $\zeta_{mean}/ \zeta_{max}$ can be in real datasets, especially those considered in your experiments? Because I think such fraction can be very small in practice, which is also acknowledged in your Section 2.1. So the regime in which MS provides benefits is in fact quite limited.





[1] https://arxiv.org/pdf/2006.09365.pdf

**Questions:**

Please see weaknesses.

---

> ### Author Response · Authors · 2023-11-15
> **Reply to Reviewer p51w**
>
> Thank you for your helpful feedback.
>
> **About Weaknesses 1 and 2**.
>
> First of all, we want to emphasize that the most critical difference of our study from CCLIP paper [1] is the improvement of the optimization error; ***our obtained error is $\delta$ times smaller than that of CCLIP in the best case*** (i.e., $\zeta_\mathrm{max} \approx \zeta_\mathrm{mean}$). Since $\delta$ is often a small value in distributed learning systems, this improvement is quite important from both a theoretical and a practical point of view. The condition $\delta \leq \zeta_\mathrm{mean} / \zeta_\mathrm{max}$ holds empirically in our experiments (please see the comments to ``About Weakness 3'').
>
> Below, we summarize the main differences between our Momentum Screening (MS) and CCLIP [1] that are mentioned in our paper.
> + Algorithmically, as shown in Section 3, our algorithm relies on the screening technique rather than clipping, which is critical for our theoretical analysis.
> + Theoretically, as described in Section 1 (Main contribution and Related work) and Section 4 (Remark 2), the optimization error can be $\delta$ times better than that of CCLIP.
> + Empirically, as provided in Section 7, we demonstrated the consistent superiority of our method over CCLIP in various numerical experiments.
>
> Also, the derivation and the results of the aggregation error bound (Proposition 2) and the momentum diameter bound (Proposition 3) under the uniform gradient heterogeneity condition is the key technical part of our analysis, and is clearly different from the analysis of CCLIP, where only the mean gradient heterogeneity condition is assumed and the aggregation error bound is $\delta$ times worse than ours due to the clipping bias.
>
> **About Weakness 3**.
>
> First, the expression $\delta \leq (\zeta_\mathrm{max}/\zeta_\mathrm{mean})^2$ was fixed in the revised paper as $\delta \leq (\zeta_\mathrm{mean}/\zeta_\mathrm{max})^2$. The typos do not affect the proofs given in Sections A and B.  We thank you for pointing this out and apologize for these typos.
>
> Second, regarding your concern, we recognize that empirical validation of the condition $\delta \leq (\zeta_\mathrm{mean}/\zeta_\mathrm{max})^2$ is critical in our study. After reading your review, we examined the empirical values of $\zeta_\mathrm{mean}/\zeta_\mathrm{max}$ on the settings of Section 6 and additionally on Fed-EMNIST, which is known as a FL dataset for a realistic situation. From Figure 14 in Section E of the revision paper (p.36), we can see that ***$\zeta_\mathrm{mean}/\zeta_\mathrm{max}$ was in $0.5 \sim 0.93$ on MNIST and CIFAR10, and was in $0.34 \sim 0.70$ on Fed-EMNIST***. Thus, we conclude that the condition ***$\delta \leq (\zeta_\mathrm{mean}/\zeta_\mathrm{max})^2$ is practical enough and the benefits of MS are not so limited***.
>
> We would be very happy if your concerns were addressed and the score would be raised.

---

### Comment · Area_Chair_1stn · 2023-11-20
**Some questions from AC**

The AC has been reading the papers and the discussions between authors and reviewers recently. Currently, the AC has the following questions for this paper, can the authors also provide some comments?

(1). Assumption 3 and 4 are strong assumptions that are not adopted in the existing works like CClip. Therefore, MS seems to achieve the proposed improvement by much stronger assumptions.

(2). Assumption 3 seems redundant as the boundedness of Assumption 4 is stronger than the sub-Gaussian assumed in Assumption 3.

(3). As CClip algorithm already achieves an $O(\delta\zeta_{mean}^2)$ complexity, hence there seem to be an effective regime within which the lower bound holds. That is, there should be some $\delta_0>0$ such that the derived lower bound holds for $\delta<\delta_0$. However, this fact is not reflected in Theorem 2.

---

> ### Author Response · Authors · 2023-11-21
> **Reply to AC**
>
> Thank you for your careful reading and important questions.
>
> **About Question (1)**:
>
> **Assumption 3**:
> As you said, Assumption 3 is actually stronger than the standard Stochastic Gradient Variance Boundedness (SGVB) assumption, which requires $\mathbb{E}\\|g_i - \nabla f_i(x)\\|^2 \leq \sigma^2$ for minibatch stochastic gradient $g_i$ at $x$ of worker $i$. However, this stronger requirement comes from the fact that ***our theory derives a high-probability bound instead of an expectation bound***, which is a much stronger result. ***Assumption 3 is a standard assumption for deriving a high-probability bound*** in the stochastic optimization literature not only in the context of the Byzantine robust optimization; in general, deriving an expectation bound requires SGVB, and deriving a high-probability bound requires Assumption 3 (see, for example, Section 2.3 of [1]).
>
> [1] Jin et al., 2019: On Nonconvex Optimization for Machine Learning: Gradients,
> Stochasticity, and Saddle Points.
>
> **Assumption 4**:
> Assumption 4 has very little effect on our theoretical results even if $G$ can be much larger than $\sigma$ in Assumption 3. In fact, in Theorem 1, ***$G$ depends only log log order on the non-dominant terms of the convergence rate*** (and $G$ never depends on the final optimization error $O(\delta^2\zeta_\mathrm{max}^2)$), as described immediately after Assumption 4 in Section 2. Thus, ***Assumption 4 is not restrictive at all***.
>
>
> **About Question (2)**:
>
> This is not true. Generally, $\sigma^2$ in Assumption 3 can be much smaller than $G^2$ in Assumption 4, since $\sigma^2$ approaches to zero as the minibatch size goes to $\infty$. In Theorem 1, $\sigma^2$ in Assumption 3 arises explicitely in the convergence rate. In contrast, as mentioned in the answer to Question (1), $G$ in Assumption 4 depends only log log order on the non-dominant terms of the convergence rate. Thus, Assumption 3 is not redundant, since replacing $\sigma^2$ with $G^2$ gives a much worse bound.
>
> **About Question (3)**:
>
> First, we would like to clarify your concern. We think your concern is why the upper bound $O(\delta \zeta_\mathrm{mean}^2)$ of CClip seems to be better than our lower bound $\Omega(\delta^2 \zeta_\mathrm{max}^2)$ for  some $\delta \in (0, 0.5)$ when $\zeta_\mathrm{mean} < \zeta_\mathrm{max}$. Is our understanding correct? If so, we would like to address this important point.
>
> In fact, ***the lower bound derived in Theorem 2 holds for any $\delta \in [0, 0.5)$ for function class $\mathcal C_\mathrm{UH}(\zeta_\mathrm{max})$ and does not contradict the upper bound of CClip for any $\delta \in [0, 0.5)$***.
> This point is explained as follows.
> First, please note that the theory of CClip essentially gives an upper bound $O(\delta \zeta_\mathrm{mean}^2)$ for $\mathcal C_\mathrm{MH}(\zeta_\mathrm{mean})$.
> Using this fact, ***a trivial upper bound $O(\delta \zeta_\mathrm{max}^2)$ by applying CClip theory to
> $\\{f_i\\}\_{i \in \mathcal G}  \in \mathcal C_\mathrm{UH}(\zeta_\mathrm{max})$ ($\subset \mathcal C_\mathrm{MH}(\zeta_\mathrm{max})$) is larger than our lower bound $\Omega(\delta^2 \zeta_\mathrm{max}^2)$ in Theorem 2, for any $\delta \in [0, 0.5)$***. Please note that $\mathcal C_\mathrm{UH}(\zeta_\mathrm{max}) \not \subset \mathcal C_\mathrm{MH}(\zeta_\mathrm{mean})$ for any $\zeta_\mathrm{mean} < \zeta_\mathrm{max}$ in general, and thus the theory of CClip never guarantees the upper bound $O(\delta \zeta_\mathrm{mean}^2)$ for $\mathcal C_\mathrm{UH}(\zeta_\mathrm{max})$ for any $\zeta_\mathrm{mean} < \zeta_\mathrm{max}$. Thus, the upper bound $O(\delta \zeta_\mathrm{mean}^2)$ of CClip for $\mathcal C_\mathrm{MH}(\zeta_\mathrm{mean})$ does not contradict the lower bound in Theorem 2 for any $\delta \in [0, 0.5)$.
>
> We would be happy to clarify any further concerns and questions.

---

### Author Response · Authors · 2023-11-21
**To all reviewers**

We thank all reviewers for their valuable comments.

As mentioned in the replies to Reviewers p51w and BuTUe, we did additional experiments on the validity of the condition $\delta \leq (\zeta_\mathrm{mean}/\zeta_\mathrm{max})^2$ on Fed-EMNIST, which is a real federated learning dataset. From Figure 14 in Section E of the revision paper (p.36), we found that $\zeta_\mathrm{mean}/\zeta_\mathrm{max}$ was in 0.34 ~ 0.7 and not so small, and thus the condition can be practical enough.

We then also performed accuracy comparisons of the proposed method with the existing methods against the five attacks described in Section 6 for the case of $\delta = 0.1$ on Fed-EMNIST. From Table 3 in Section F of the revision paper, we found that the proposed method still consistently outperformed the other methods in terms of the worst best test accuracy against the five attacks.

We hope that if these results are helpful in addressing your concerns.

---

### Meta-Review · Area_Chair_1stn · 2023-12-04

**Metareview:**

This paper studies the nonconvex federated learning problem with Byzantine workers. Let the fraction of Byzantine workers be $\delta$, a momentum SGD enhanced with a screening test is proposed, and an $O(\delta^2\zeta_{max}^2)$ error can be achieved, in addition, an $\Omega(\delta^2\zeta_max^2)$ lower bound has been provided. When $\delta$ is a small quantity and $\zeta_{max}$ is not much larger than $\zeta_{mean}$, then this is a large improvement compared to the $O(\delta^2\zeta_{mean})$. However, the stronger result is based on the stronger light-tailed noise assumption, without which the high probability bound and the screening would not be valid. Moreover, the convergence rate in terms of the number of steps might not be optimal. Overall, the screening technique, the lower bound, and the improved final accuracy are all valid contribution in this paper.

As the reviewers are having a split reviews for this paper, the AC also read this paper to understand the concerns from the negative, which is about the difference of the algorithm and the results compared with the CCLIP method (Karimireddy et al. (2022)). The AC thinks that the authors' response to the negative reviewer is valid (which is not responded by reviewer), and the authors also clarified a few additional questions from the AC. As a consequence, the AC decides to override the negative review and propose an accept to this paper.

**Justification For Why Not Higher Score:**

Stronger assumption than before, and the convergence rate in terms of the number of steps might not be optimal.

**Justification For Why Not Lower Score:**

The paper does have some theoretical merit in proposing the screening technique, the lower bound, and the improved final accuracy.

---

### Decision · Program_Chairs · 2024-01-16

Accept (poster)